# Dephosphorylation of the HIV-1 restriction factor SAMHD1 is mediated by PP2A-B55α holoenzymes during mitotic exit

Kerstin Schott[1], Nina V. Fuchs[1], Rita Derua[2,3], Bijan Mahboubi[4], Esther Schnellbächer[1], Janna Seifried[1], Christiane Tondera[1], Heike Schmitz[1], Caitlin Shepard[4], Alberto Brandariz-Nuñez [5], Felipe Diaz-Griffero[5], Andreas Reuter[6], Baek Kim[4,7], Veerle Janssens [2] & Renate König[1,8,9]

SAMHD1 is a critical restriction factor for HIV-1 in non-cycling cells and its antiviral activity is regulated by T592 phosphorylation. Here, we show that SAMHD1 dephosphorylation at T592 is controlled during the cell cycle, occurring during $M/G_1$ transition in proliferating cells. Using several complementary proteomics and biochemical approaches, we identify the phosphatase PP2A-B55α responsible for rendering SAMHD1 antivirally active. SAMHD1 is specifically targeted by PP2A-B55α holoenzymes during mitotic exit, in line with observations that PP2A-B55α is a key mitotic exit phosphatase in mammalian cells. Strikingly, as HeLa or activated primary CD4$^+$ T cells enter the $G_1$ phase, pronounced reduction of RT products is observed upon HIV-1 infection dependent on the presence of dephosphorylated SAMHD1. Moreover, PP2A controls SAMHD1 pT592 level in non-cycling monocyte-derived macrophages (MDMs). Thus, the PP2A-B55α holoenzyme is a key regulator to switch on the antiviral activity of SAMHD1.

[1] Host-Pathogen Interactions, Paul-Ehrlich-Institut, 63225 Langen, Germany. [2] Department of Cellular and Molecular Medicine, Laboratory of Protein Phosphorylation and Proteomics, KU Leuven, 3000 Leuven, Belgium. [3] Facility for Systems Biology based Mass Spectrometry (SYBIOMA), KU Leuven, 3000 Leuven, Belgium. [4] Center for Drug Discovery, Department of Pediatrics, Emory University, Children's Healthcare of Atlanta, Atlanta, GA 30322, USA. [5] Department of Microbiology and Immunology, Albert Einstein College of Medicine, Bronx, NY 10461, USA. [6] Division of Allergology, Paul-Ehrlich-Institut, 63225 Langen, Germany. [7] Department of Pharmacy, Kyung-Hee University, 2447 Seoul, South Korea. [8] Immunity and Pathogenesis Program, Sanford Burnham Prebys Medical Discovery Institute, La Jolla, CA 92037, USA. [9] German Center for Infection Research (DZIF), 63225 Langen, Germany. These authors contributed equally: Nina V. Fuchs, Rita Derua. Correspondence and requests for materials should be addressed to R.Kön. (email: Renate.Koenig@pei.de)

Sterile α motif (SAM) domain and HD domain-containing protein 1 (SAMHD1) is a $Mg^{2+}$-dependent triphosphohydrolase (dNTPase) converting deoxynucleoside triphosphates (dNTPs) into deoxynucleosides and inorganic triphosphates[1]. Besides the dNTPase function, SAMHD1 binds to single-stranded nucleic acids[2,3] and is proposed to exert nuclease activity[4–6], a function which is heavily debated[3,7,8].

Mutations in *SAMHD1* cause the hereditary autoimmune disease Aicardi-Goutières syndrome (AGS), associated with elevated production of interferon (IFN) α[9]. Moreover, SAMHD1 is frequently mutated in a variety of cancer types, such as chronic lymphocytic leukemia (CLL) and colorectal cancer[10,11]. Importantly, SAMHD1 restricts a diverse set of DNA and retroviruses[12–15]: In particular, human immunodeficiency virus (HIV)-1 is restricted at an early replication step in non-cycling myeloid cells and resting $CD4^+$ T cells[16–19]. As a potent dNTPase, SAMHD1 efficiently reduces cellular dNTP levels in non-cycling cells below those required to support HIV-1 reverse transcription (RT)[1,20]. Furthermore, SAMHD1's RNase activity was proposed to mediate HIV-1 restriction[5]; it is, however, unclear whether this additional enzymatic activity may be causative for HIV-1 inhibition[3,8].

Regardless of the precise restriction mechanism, SAMHD1 expression alone is not sufficient to induce a potent block of HIV-1 replication, as activated $CD4^+$ T and cycling THP-1 cells express high SAMHD1 levels, but are permissive for HIV-1 infection[16,18]. SAMHD1 is phosphorylated at threonine (T) 592 in asynchronously proliferating cells (SAMHD1 pT592), rendering it inactive against HIV-1[21–23]. SAMHD1 interacts with cyclin-dependent kinase (CDK) 1 and 2/cyclin A2 in cycling cells[21,24], in accordance with T592 as a target site for CDKs (consensus sequence: S/T-P-x-K/R, SAMHD1 motif: $^{592}$TPQK$^{595}$). How T592 phosphorylation of SAMHD1 influences its structural and enzymatic properties, tetramerization propensity[25–28] and dNTPase activity[22,23], is a matter of debate. Nevertheless, only dephosphorylated SAMHD1 at T592 is able to actively restrict HIV-1[21–24]. Remarkably, the importance of a dephosphorylated antiviral-active state of SAMHD1 has been proposed for hepatitis B virus (HBV)[15] as well, suggesting this specific post-translational modification as an important regulatory mechanism. Besides the control of SAMHD1's antiviral activity, phosphorylation at T592 has been proposed to play a novel role in promoting the resection of arrested replication forks and preventing the accumulation of single-stranded DNA (ssDNA) derived from stalled forks in the cytoplasm[29]. This reinforces the importance of both, phosphorylation and dephosphorylation at this specific residue, for diverse physiological functional states of SAMHD1.

In this report, two complementary proteomics approaches identified the serine/threonine protein phosphatase 2 A (PP2A) as the responsible phosphatase actively removing the phosphate at T592 in SAMHD1. Particularly, PP2A holoenzymes containing the regulatory subunit B55α, which is critical for substrate specificity, efficiently acted on T592 in vitro and in cells. Intriguingly, PP2A-B55α holoenzymes are responsible for dephosphorylation of SAMHD1 at T592 in proliferating cells during mitotic exit, an important transition between M and $G_1$ phase of the cell cycle. Concomitantly, we observed a rapid drop in dATP levels, suggesting either a coincidental or causative relationship between dephosphorylation and dNTPase activity. Importantly, upon entry into $G_1$ phase, HIV-1 infection led to reduction of early and late RT products in activated $CD4^+$ T and HeLa cells, depending on the presence of dephosphorylated SAMHD1. Thus, we defined the time window of PP2A activity during which SAMHD1 is rendered antivirally active. Additionally, PP2A controls SAMHD1 T592 phosphorylation in non-cycling MDMs,

important HIV-1 target cells. Furthermore, we provide evidence for PP2A involvement in the IFN-inducible dephosphorylation of SAMHD1 in MDMs.

## Results

**Cell cycle-dependent regulation of SAMHD1 pT592 level**. To characterize the cell cycle-dependent (de)phosphorylation of SAMHD1 at T592 in more detail, we synchronized HeLa cells at the $G_1$/S border using a double-thymidine block. Cell cycle-progression was monitored by immunoblotting using cyclin-specific antibodies (Fig. 1a) and by flow cytometric analysis of DNA content (Fig. 1b). Interestingly, SAMHD1 protein levels remained constant in all cell cycle phases (Fig. 1a), including S phase (0–4 h post-release). SAMHD1 phosphorylation at T592 appeared high in early S phase (0–4 h post-release)—consistent with reports of initial CDK2-dependent phosphorylation at T592[24,30]. After maximal activity of CDKs/cyclin A2, SAMHD1 phosphorylation is maintained in $G_2$/M phase (7–8 h post-release)—accompanied by maximal cyclin A2/B1 expression (Fig. 1a). However, when cells re-enter interphase, SAMHD1 phosphorylation at T592 is rapidly lost and paralleled by complete degradation of cyclins A2/B1 (starting at 10 h and maximal at 11.5 h post-release) (Fig. 1a). Moreover, we monitored dATP levels during cell cycle progression (Fig. 1c, Supplementary Fig. 1a, b). Interestingly, dATP levels remain consistently high from S to $G_2$/M phase (0–8 h post-release). However, with transition of cells into $G_1$ phase, starting at 9 h post-release (Supplementary Fig. 1b), SAMHD1 dephosphorylation (Supplementary Fig. 1a) is accompanied by a pronounced decrease in dATP levels (Fig. 1c). SAMHD1 is again phosphorylated at T592 as cells re-enter S phase (16–18 h post-release; Supplementary Fig. 1d, e) and dATP levels start to rise (13 h post-release; Fig. 1c and Supplementary Fig. 1a, b). Taken together, these results suggest the involvement of a cellular phosphatase that actively removes the T592 phosphate from SAMHD1 during M/$G_1$ transition.

**PP2A interacts with SAMHD1 in proliferating cells**. To assess which phosphatase potentially binds to and acts on SAMHD1 upon entry into $G_1$ phase of the cell cycle, we conducted two independent mass spectrometry (MS) experiments in asynchronously growing human embryonic kidney (HEK) 293T cells. We performed affinity purification of SAMHD1-associated protein complexes followed by in-gel or on-bead digestion, respectively, and subsequent MS analysis (see methods; Fig. 2a and Supplementary Fig. 2a-c). Interestingly, we identified the known phosphorylating kinase CDK1 and the PP2A Aα subunit as SAMHD1 co-purifying proteins in both MS experiments (Fig. 2a and Supplementary Fig. 2a-c), suggesting PP2A as a potential SAMHD1-interacting phosphatase. To validate the MS findings, we over-expressed FLAG-tagged SAMHD1 in HEK293T cells and identified both endogenous PP2A A and C subunits in the pull-down (Fig. 2b).

To further characterize the interaction of SAMHD1 with PP2A, we performed co-immunoprecipitations (CoIPs) with SAMHD1 mutants in HEK293T cells (Supplementary Fig. 2d). None of the tested mutants, including single exchange of amino acids essential for SAMHD1 catalytic activities, displayed an altered binding capability to PP2A A and C subunit (Supplementary Fig. 2d). Exchange of the CDK target site T592 to a phosphomimetic (T592E) or a non-phosphorylatable (T592A) residue did also not alter binding of PP2A A and C subunits, indicating that PP2A binding is also not depending on this specific motif. Moreover, a mutant of SAMHD1 (R451E) known to prevent formation of dimers and tetramers[31] influenced

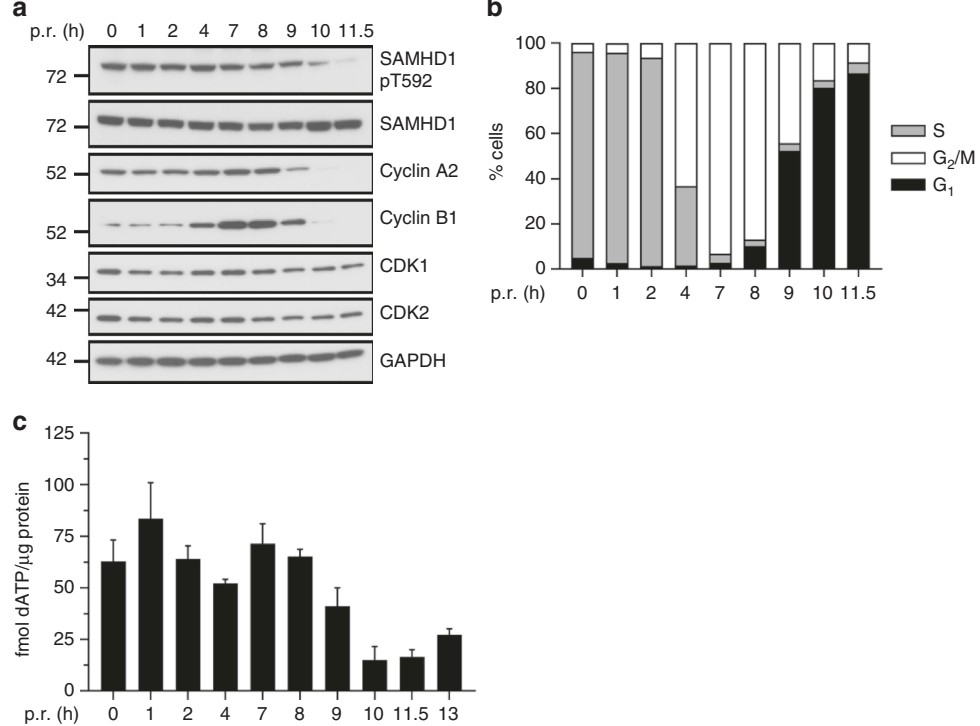

**Fig. 1** SAMHD1 (de)phosphorylation at T592 is regulated in a cell cycle dependent manner. **a**, **b** HeLa cells were arrested at the $G_1$/S border using a double-thymidine block. After the 2nd release, synchronized cells were harvested at different time points post-release (p.r.). Respective samples were split for immunoblotting (**a**) and propidium iodide (PI) staining (**b**) to determine cell cycle-phases by flow cytometry. For immunoblotting, whole-cell lysates were analyzed using antibodies specific to the indicated proteins. Data shown are representative of two independent experiments. **c** HeLa cells were arrested at the $G_1$/S border and harvested at different time points as described in Fig. 1a. Respective samples were split for dATP measurement, immunoblotting (Supplementary Fig. 1a, c) and PI staining (Supplementary Fig. 1b). dATP was quantified by single nucleotide incorporation assay and normalized by protein content of the corresponding lysate. Data shown represent the mean ± SD of two independent experiments

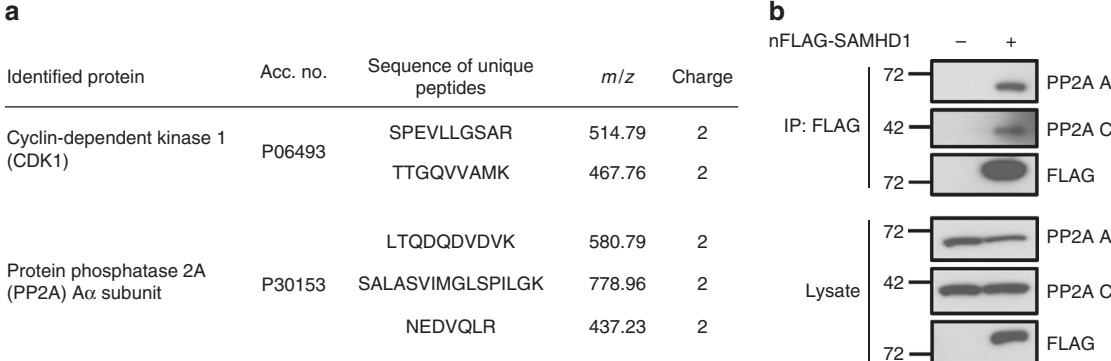

**Fig. 2** SAMHD1 interacts with endogenous PP2A in cycling cells. **a** Identification of PP2A Aα subunit and CDK1 as SAMHD1-interacting proteins using MS-based interactomics. Anti-GFP trapping of the same amount of lysates from GFP or GFP-SAMHD1 overexpressing cells was executed. The traps were subjected to on-bead trypsin digestion followed by nano-LC-MS/MS analysis for identification of interacting proteins. PP2A Aα subunit and CDK1 were present on the GFP-SAMHD1 trap and not on the GFP trap, their identification being based on the presence of 3, respectively 2 unique peptides. **b** SAMHD1 interacts with endogenous PP2A A and C subunits. HEK293T cells were transfected with constructs expressing N-terminally FLAG-tagged SAMHD1 or empty vector as a negative control. 48 h post-transfection, cells were harvested, lysed and co-immunoprecipitations (CoIPs) performed using anti-FLAG-coated agarose beads. Proteins were analyzed by immunoblotting using antibodies specific to the indicated proteins. Data shown are representative of three independent experiments

binding to PP2A A and C subunits only to a minor extent (Supplementary Fig. 2e), suggesting that residues responsible for PP2A binding might be accessible in all structural states of SAMHD1.

**SAMHD1 pT592 is dephosphorylated by PP2A in vitro.** Although members of both the PP2A and the protein

phosphatase 1 (PP1) families are known to counteract CDK1-mediated phosphorylations during mitosis in mammalian cells[32], the MS and subsequent CoIP experiments rather suggested PP2A as the most plausible SAMHD1 phosphatase.

To clarify which phosphatase acts on SAMHD1 at T592, we performed in vitro-dephosphorylation assays using recombinant SAMHD1 purified from Sf9 cells that proved to be already significantly phosphorylated on T592[33]. For the

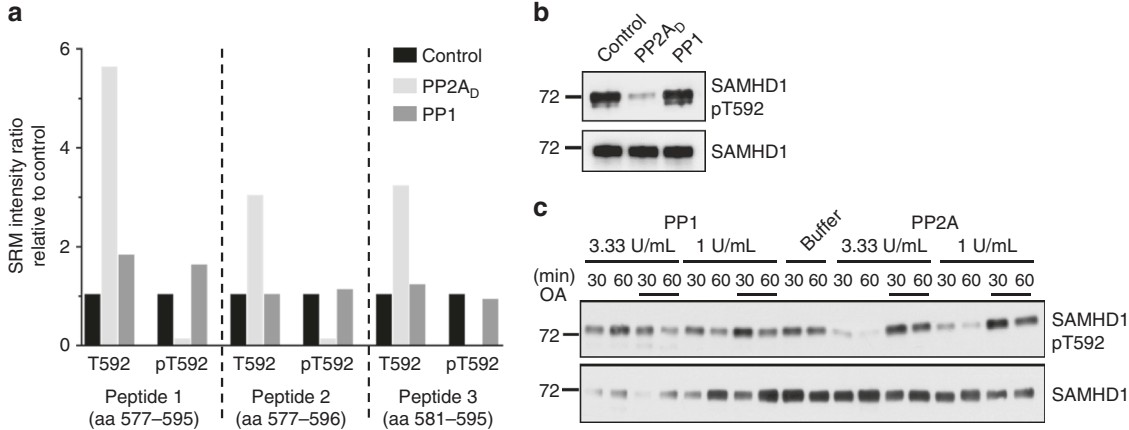

**Fig. 3** PP2A, not PP1, is able to dephosphorylate SAMHD1 pT592 in vitro. **a** Targeted nano-LC-MS/MS analysis reveals PP2A as responsible phosphatase acting on SAMHD1 pT592. 1 μg of recombinant SAMHD1 was incubated with equilibrated units of PP2A$_D$ and PP1, or buffer alone (control) for 1 h at 30 °C. After stopping the reaction by addition of 5 × SDS-PAGE sample buffer, the 1D-gel piece containing SAMHD1 was subjected to trypsin digestion followed by targeted nano-LC-MS/MS of the resulting peptides. Relative quantification of phosphorylated and non-phosphorylated peptides encompassing T592 in different conditions (treatment with PP2A$_D$ or PP1 compared to buffer only (control)) is displayed by means of their SRM intensity ratios. **b** Validation of MS data by immunoblot analysis with anti-SAMHD1 pT592 antibodies, executed on the same samples as described in Fig. 3a. The data are representative of three independent experiments. **c** Time- and concentration-dependency of SAMHD1 pT592 dephosphorylation by PP2A$_D$. In vitro-dephosphorylation of recombinant SAMHD1 was executed with the indicated units of PP1 or PP2A$_D$ for the indicated time points. When appropriate, 50 nM OA was added to the phosphatases in a 10 min-pre-incubation step, before addition of the substrate, and dephosphorylation for the indicated times. SAMHD1 (de) phosphorylation was monitored by immunoblotting of samples with SAMHD1 pT592-specific antibodies

dephosphorylation, we used equilibrated units of a purified PP2A dimer (PP2A$_D$; = PP2A A and C subunit) and a purified PP1 enzyme. After in vitro-dephosphorylation, the assay mixture was resolved by 1D-SDS-PAGE and the gel pieces containing SAMHD1 were digested with trypsin. The resulting peptide mixture was subjected to targeted LC-MS/MS (tandem mass spectrometry), followed by relative quantification analysis of phosphorylated and non-phosphorylated peptides encompassing T592 (Fig. 3a). The intensities of selected reaction monitoring (SRM) transitions corresponding to phosphorylated peptides were clearly diminished upon PP2A treatment in comparison to treatment with PP1 or buffer only. Concomitantly, the intensities of SRM transitions corresponding to the non-phosphorylated counterparts were higher upon PP2A treatment in comparison to treatment with PP1 or buffer only (Fig. 3a). From these intensity values, an average T592 phosphorylation stoichiometry of 76% could be calculated for control SAMHD1, whereas after PP2A treatment the average T592 phosphorylation stoichiometry goes down to 5% (Supplementary Table 1).

Immunoblot analysis further validated these MS data, revealing significantly decreased pT592 immune-reactivity, only upon treatment with PP2A$_D$ (Fig. 3b). Additionally, SAMHD1 pT592 dephosphorylation by PP2A$_D$ was time- and concentration-dependent, and could be completely inhibited by PP2A-selective concentrations of okadaic acid (OA), a pharmacological inhibitor of PP2A-like phosphatases[34] (Fig. 3c). Thus, our in vitro-dephosphorylation assays clearly exclude PP1 as a SAMHD1 phosphatase, and provide further support for PP2A as the likely enzyme dephosphorylating pT592.

**PP2A-B55α interacts with and dephosphorylates SAMHD1 at T592.** Active PP2A holoenzymes are heterotrimers consisting of a scaffolding A subunit (2 isoforms: Aα/β; *PPP2R1A/B*), a catalytic C subunit (2 isoforms: Cα/β; *PPP2CA/B*) and a regulatory B-type subunit, which modulates substrate specificity as well as the temporal and spatial activity of the enzyme in cells[35]. PP2A B-type subunits are encoded by 15 different genes—resulting in 23 isoforms, which are classified into four different families: B (B55/

PR55; *PPP2R2*), B′ (B56/PR61; *PPP2R5*), B″ (PR72; *PPP2R3*) and B‴ (striatins; *STRN*)[34]. Thus, we set out to identify the particular PP2A holoenzyme(s) specifically interacting with and dephosphorylating SAMHD1 at the M/G$_1$ transition in cycling cells.

First, we isolated specific PP2A holoenzymes by expressing a large set of glutathione S-transferase (GST)-tagged B-type subunits in HEK293T cells and subsequently analyzed co-purification of FLAG-tagged SAMHD1. Interestingly, SAMHD1 was found enriched after pull-down of the PP2A-B55α holoenzyme, whereas no appreciable interaction could be observed with any other B-type subunit tested (Fig. 4a). At this point, we cannot exclude binding of other subtypes that may interact with SAMHD1 with lower affinity and could possibly target other phosphorylation sites. However, the reciprocal pull-down of SAMHD1 confirmed the interaction with PP2A-B55α trimers (Fig. 4b). Moreover, FLAG-tagged SAMHD1 co-purified with endogenously expressed PP2A-B55α holoenzymes in proliferating HEK293T cells (Fig. 4c). Endogenous SAMHD1 immunoprecipitated from proliferating THP-1 cells interacted with PP2A B55α subunit (Fig. 4d), thereby validating the interaction in the native-like environment at endogenous levels of both proteins. In line with our CoIP results, loss of SAMHD1 T592 phosphorylation during cell cycle-progression resembles PRC1 pT481 dephosphorylation (Supplementary Fig. 1c, f), a known target of B55α-containing PP2A holoenzymes[36].

Known PP2A-B55α substrates share a common feature, a bipartite polybasic recognition determinant surrounding the CDK site in question[37]. Indeed, we identified basic patches surrounding the T592 site in human SAMHD1 that share similarity to the reported recognition determinant for B55 substrates[37] (Fig. 4e, highlighted in black). To directly test if SAMHD1 might be a substrate of B55 and assess the specificity of interaction, we compared basic residues in the C-terminal region of human and murine (isoform 1) SAMHD1 surrounding T592. Isoform 1 of murine SAMHD1 contains less basic residues (Fig. 4e, highlighted in gray), which correlated with an almost complete loss of interaction with PP2A (Fig. 4f; Hs vs Mm wt). Removing basic residues in human SAMHD1 reduced (Fig. 4e, f; Hs·Mm),

whereas introduction of basic residues into murine SAMHD1 markedly increased interaction with PP2A after pull-down (Fig. 4e, f; Mm·Hs), demonstrating key sites in SAMHD1 responsible for binding to PP2A-B55α holoenzymes. Taken together, these experiments demonstrate that PP2A-B55α holoenzymes specifically interact with SAMHD1 in asynchronously proliferating cells.

Next, we isolated PP2A-B55α holoenzymes from GFP-B55α-expressing HEK293T cells by a GFP-trapping approach, and used this PP2A trimer to dephosphorylate SAMHD1 pT592 in vitro. Again, a time- and concentration-dependent pT592

dephosphorylation was observed that was completely inhibited by addition of OA (Fig. 5a). When compared with in vitro-dephosphorylation by PP2A$_D$, dephosphorylation by PP2A-B55α trimer proved at least as efficient (Fig. 5b), consistent with the reported preference of B55-type regulatory subunits to direct PP2A activity to phosphoserine-proline or phosphothreonine-proline sites[37,38].

To further validate the specificity of the PP2A-B55α holoenzyme, recombinant SAMHD1 was incubated with respective holoenzymes of the PP2A B56/B′ and PR72/B″ subunit families. PP2A-B56α, PP2A-B56β, and PP2A-PR72 holoenzymes which

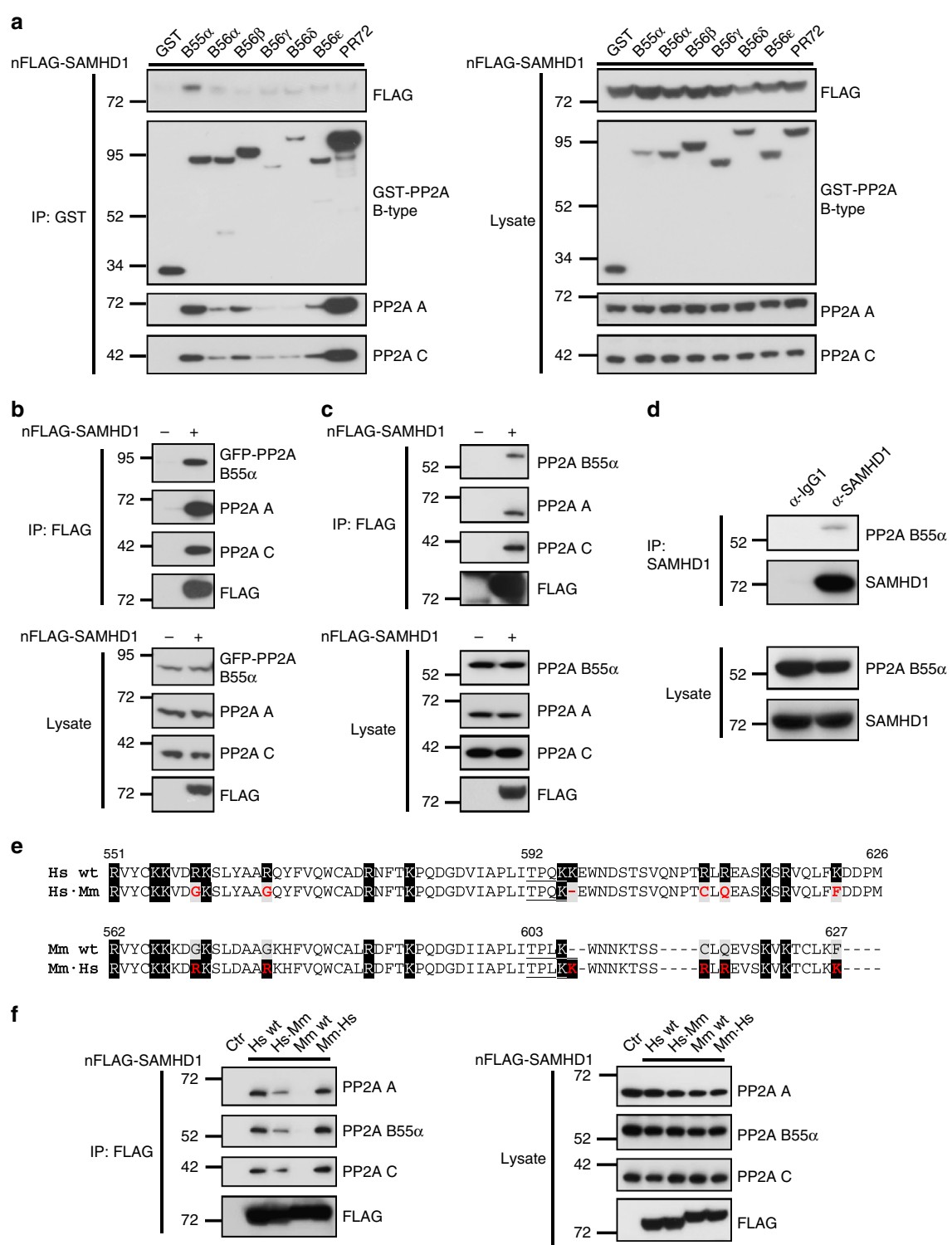

did not bind to SAMHD1 in cells (Fig. 4a) were also not able to remove T592 phosphorylation in vitro (Fig. 5c; Supplementary Fig. 4e). Taken together, our results not only support a specific interaction of PP2A-B55α holoenzymes with SAMHD1 in cells, but also emphasize specific SAMHD1 pT592 dephosphorylation by this PP2A trimer.

**Silencing of PP2A-B55α increases SAMHD1 T592 phosphorylation.** To underscore the involvement of PP2A in SAMHD1 pT592 dephosphorylation, we treated HeLa cells with low, PP2A-selective concentrations (nM range) of OA[34] for 8 h. Increased SAMHD1 phosphorylation at T592 was observed (Fig. 5d), suggesting that the inhibition of PP2A by OA leads to accumulation of phosphorylated SAMHD1.

To provide further support, we transfected several siRNAs that were previously shown to be on-target for single or all subunits of PP2A-B55α trimers[39] into HeLa cells. 52 h post-transfection, protein levels of PP2A subunits were successfully decreased as confirmed by immunoblotting (Fig. 5e). As reported before, downregulation of PP2A Aα or Cα subunit leads to co-depletion of other subunits, including B55α[39]. Upon PP2A single subunit or triple knock-down, phosphorylation of endogenous SAMHD1 was increased compared to control siRNA. Intriguingly, already single knockdown of the PP2A B55α subunit led to an increase of SAMHD1 T592 phosphorylation (Fig. 5e), indicating that PP2A-B55α holoenzymes are mediating SAMHD1 dephosphorylation at this specific residue in cells. Surprisingly, we observed overall SAMHD1 protein levels decreasing after PP2A knockdown (Fig. 5e), suggesting that hyperphosphorylated SAMHD1 might be degraded by an unknown mechanism.

**PP2A-B55α controls SAMHD1 pT592 level in non-cycling MDMs.** We next analyzed whether phosphorylation of SAMHD1 at T592 was also controlled by PP2A in non-proliferating primary human HIV-1 target cells, namely MDMs (Supplementary Fig. 3a). As a consequence of PP2A inhibition with a PP2A-selective concentration of OA (2 nM), SAMHD1 phosphorylation at T592 increased over 8 h in different donors (Fig. 5f). This indicates that PP2A is involved in keeping T592 phosphorylation in MDMs in balance as well. Interestingly, we observed strong donor-dependent differences in steady-state SAMHD1 phosphorylation levels at T592, which correlated with minichromosome maintenance protein 2 (MCM2) expression (Fig. 5f and Supplementary Fig. 3b). As MCM2 is not expressed in $G_0$[40], it can be used as a marker to track conversion of MDMs from the quiescent state ($G_0$) to $G_1$ phase. Interestingly, these varying levels of SAMHD1 T592 phosphorylation in different donors correlated with susceptibility to HIV-1 infection (Supplementary Fig. 3c, d). Nevertheless, in all observed donors, SAMHD1 phosphorylation at T592 increased upon inhibition of PP2A (Fig. 5f; Supplementary Fig. 3e). To prove that indeed PP2A-B55α trimers are responsible for dephosphorylating SAMHD1 in MDMs, same as in dividing cells (Fig. 5e), we silenced the PP2A B55α subunit in MDMs (Fig. 5g; Supplementary Fig. 3f). Highly efficient silencing of B55α mRNA (Fig. 5g, right panel) translates in apparent, but modest down-regulation of B55α subunit protein expression (Fig. 5g, left panel), but remarkably, T592 phosphorylation of endogenous SAMHD1 was increased compared to control siRNA in MDMs (Fig. 5g, left panel; Supplementary Fig. 3f).

To summarize, employing the phosphatase inhibitor OA and siRNA-mediated silencing of PP2A, we observed an increase in SAMHD1 phosphorylation at T592 in cycling HeLa cells (Fig. 5d, e) as well as in differentiated MDMs (Fig. 5f, g; Supplementary Fig. 3e, f). Taken together, our results demonstrate that PP2A is important for controlling SAMHD1 phosphorylation at T592, in both cycling and differentiated cells.

**Upregulation of B55α subunit in MDMs upon IFN treatment.** Previous reports have shown that SAMHD1's antiviral activity is regulated by type I IFN. SAMHD1 phosphorylation at T592 is decreasing in IFN-treated MDMs and monocyte-derived dendritic cells[21]. It was speculated that IFNs may induce an unknown phosphatase activity[21]. We hypothesized that specific subunits of PP2A may be differentially induced by IFNs. Therefore, we analyzed mRNA levels of various regulatory B-type subunits after treatment of MDMs with type I and II IFNs or upon infection with Sendai virus as a viral IFN inducer (Supplementary Fig. 4a, c). Primers used for quantitative real-time reverse transcription PCR (RT-qPCR) were isoform-specific, enabling us to discriminate between different PP2A B-type subunits and isoforms (Supplementary Fig. 4d). Tested B-type subunits were not detectable (B56δ) or not inducible by IFNs or Sendai virus (B56γ and ε) in MDMs, except PP2A B55α and to lower extent B56β subunit

**Fig. 4** PP2A-B55α holoenzymes interact with SAMHD1 in cycling cells. **a** SAMHD1 interacts with PP2A-B55α holoenzymes. HEK293T cells were co-transfected with constructs expressing N-terminally FLAG-tagged SAMHD1 (3.5 μg) and different N-terminally GST-tagged PP2A B-type subunits (7.5 μg). 48 h post-transfection, cells were harvested, lysed and CoIPs performed using anti-GST-coated magnetic beads. Proteins were analyzed by immunoblotting using antibodies specific to the indicated proteins. Data shown are representative of three independent experiments. **b** Reciprocal CoIP experiment. HEK293T cells were co-transfected with constructs expressing N-terminally FLAG-tagged SAMHD1 (5.5 μg) and GFP-tagged PP2A B55α subunit (5.5 μg). Sole transfection of the expression plasmid encoding GFP-tagged PP2A B55α subunit was used as a negative control. 48 h post-transfection, cells were harvested, lysed and CoIPs performed using anti-FLAG-coated agarose beads. Proteins were analyzed by immunoblotting using antibodies specific to the indicated proteins. Data shown are representative of three independent experiments. **c** SAMHD1 interacts with endogenous PP2A B55α subunit in HEK293T cells. Cells were transfected with constructs expressing N-terminally FLAG-tagged SAMHD1 or empty vector as a negative control. 48 h post-transfection, cells were harvested, lysed and CoIPs performed using anti-FLAG-coated agarose beads. Proteins were analyzed by immunoblotting using antibodies specific to the indicated proteins. Data shown are representative of two independent experiments. **d** Endogenously expressed SAMHD1 interacts with endogenous PP2A B55α subunit in monocytic THP-1 cells. Cycling THP-1 cells were harvested (2 × 10^7 cells/sample), lysed and CoIPs performed using anti-IgG1/SAMHD1-coated sepharose beads. Proteins were analyzed by immunoblotting using antibodies specific to the indicated proteins. Data shown are representative of two independent experiments. **e** Comparison of basic amino acids in the C-terminal region of human (Hs) and murine (isoform 1; Mm) SAMHD1. Alignment of SAMHD1 protein sequences was generated with ClustalW. Basic residues are highlighted in black, while non-basic residues at the corresponding positions are highlighted in gray. Introduction of non-basic residues into human SAMHD1 (Hs•Mm) and, vice versa, basic residues into murine SAMHD1 (Mm•Hs) are marked in red. **f** SAMHD1 interacts with PP2A-B55α holoenzymes through basic amino acids. HEK293T cells were transfected with constructs expressing N-terminally FLAG-tagged human/murine (isoform 1) SAMHD1 or mutants with substituted/introduced basic residues. Empty vector was transfected as a negative control. 48 h post-transfection, cells were harvested, lysed and CoIPs performed using anti-FLAG-coated agarose beads. Proteins were analyzed by immunoblotting using antibodies specific to the indicated proteins. Data shown are representative of three independent experiments

(Supplementary Fig. 4a, c). The in vitro-dephosphorylation assay clearly excluded B56β as a relevant B-type subunit for dephosphorylating T592 (Supplementary Fig. 4e), validating our previous findings for B55α as the only relevant subunit (Fig. 4 and Fig. 5). Moreover, mRNA levels of the PP2A B55α subunit were strongly induced by IFNs and Sendai virus 8 h and 24 h post-treatment, with maximal induction (up to 15.5-fold) through IFNβ (Supplementary Fig. 4a). In concordance, PP2A B55α subunit protein levels were upregulated after 24 h upon type I or II IFN induction (Supplementary Fig. 4b). Strikingly, B55α upregulation correlated with significant SAMHD1 dephosphorylation at T592 in MDMs (Supplementary Fig. 4b, left panel). In contrast to differentiated MDMs, IFN treatment of cycling HEK293T cells resulted in an upregulation

of total SAMHD1 levels, which was paralleled by an increase in T592 phosphorylation and correlated with decreasing PP2A B55α subunit levels (Supplementary Fig. 4f), suggesting a cell type-specific regulation of SAMHD1 T592 phosphorylation through IFN. Taken together, our findings on IFN inducibility of B55α in MDMs provide further support for a physiological role of the PP2A-B55α holoenzyme as the responsible SAMHD1 pT592 phosphatase.

**PP2A-B55α removes pT592 in SAMHD1 during mitotic exit.** Interestingly, PP2A-B55α holoenzymes are established regulators of mitotic exit in cycling cells[39], entirely in line with the observation that SAMHD1 loses T592 phosphorylation during $M/G_1$ transition (Fig. 1a). To characterize SAMHD1 dephosphorylation

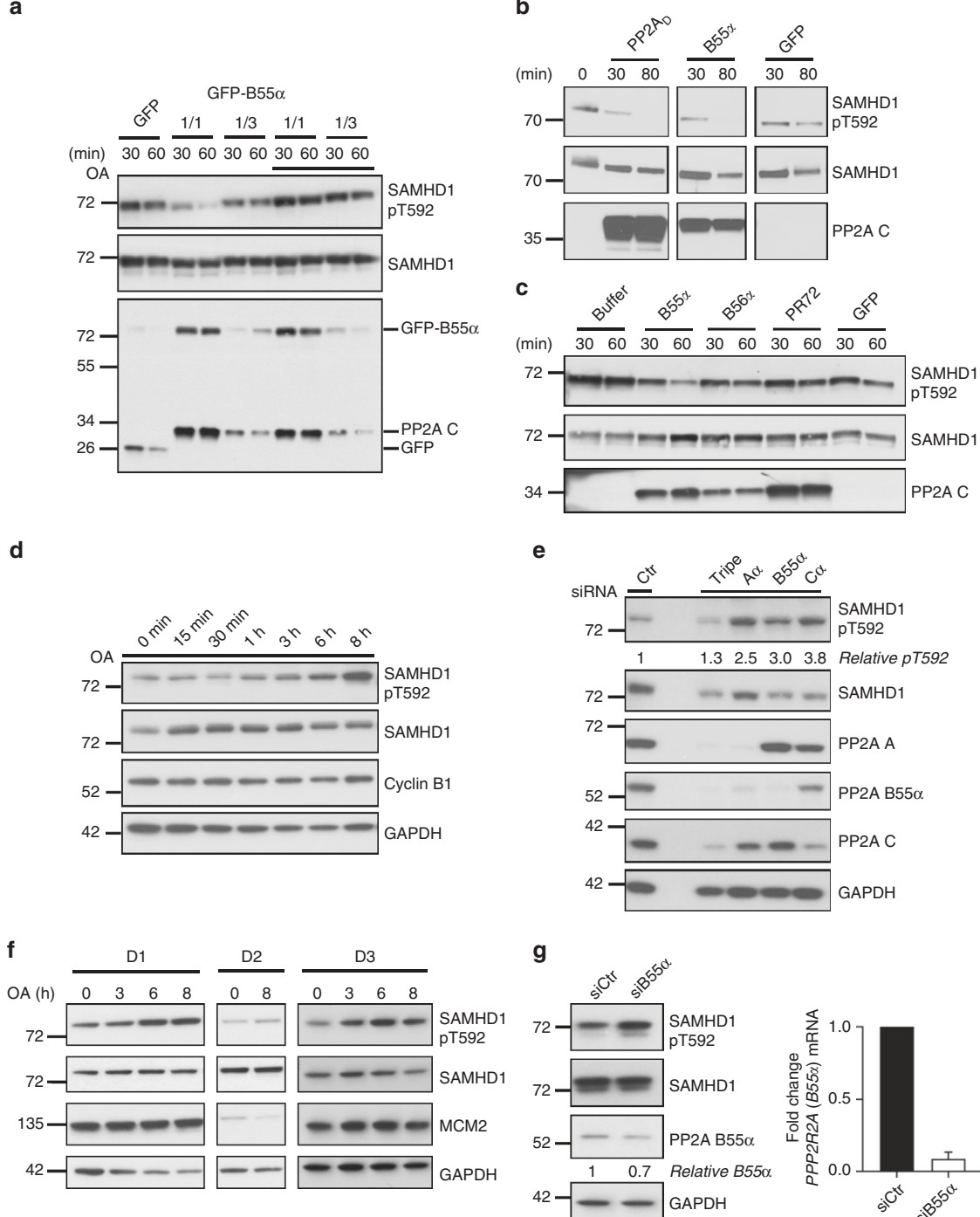

at T592 at $M/G_1$ transition in yet more detail, we chemically induced exit from mitosis in HeLa cells[39]. Using this assay, we were able to monitor dephosphorylation of CDK substrates (including SAMHD1), which is taking place during a brief time window from onset of anaphase until reformation of the nuclear envelope. HeLa cells were arrested in metaphase using nocodazole and MG-132 (proteasome inhibitor) and forced to exit mitosis by adding flavopiridol (CDK1 inhibitor) in presence of MG-132. Dephosphorylation of CDK substrates was detected at different time points after induction using an antibody specific to phosphorylated threonine (pT) within the context of the CDK target sequence pTPXK (Supplementary Fig. 5a). As expected, overall CDK substrate phosphorylation dropped to $9.3 \pm SD$ of 1.2% ($n = 3$) within 18 min after flavopiridol addition (Supplementary Fig. 5a). Phosphorylation of SAMHD1 T592 decreased similarly to $27.3 \pm 8.5\%$ ($n = 3$) within 18 min (Supplementary Fig. 5b),

demonstrating that SAMHD1 is specifically dephosphorylated during mitotic exit. Mitotic exit also resulted in a decrease in dATP levels within only 18 min (Supplementary Fig. 5c, d), which is in concordance with the drop in dATP in cells transitioning into $G_1$ phase as observed in Fig. 1c. Notably, the assay is performed while MG-132 is constantly present (as indicated by constant expression of cyclin B1; Supplementary Fig. 5a, b, d), therefore degradation of other proteins involved in dNTP metabolism are most likely not responsible for this decrease in dATP levels. In line with the requirement for PP2A-B55α for timely CDK1 substrate dephosphorylation[39], siRNA-mediated depletion of PP2A B55α subunit, delayed dephosphorylation of overall CDK substrates compared to control cells (reduction of pT signal to $28 \pm 14.1\%$ for siCtr compared to $40 \pm 15.6\%$ for siB55α; $n = 2$) (Supplementary Fig. 5e). Interestingly, also dephosphorylation of SAMHD1 at T592 was delayed compared to control cells

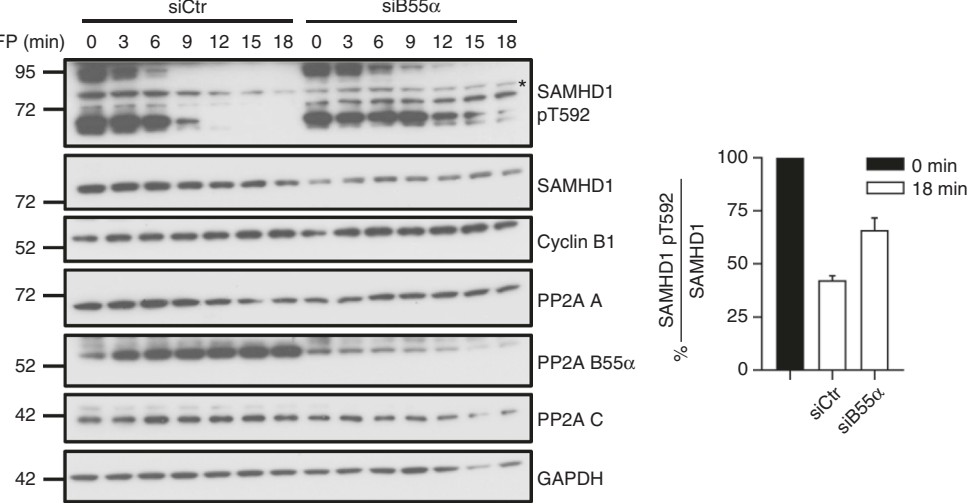

**Fig. 6** SAMHD1 is dephosphorylated at T592 by PP2A-B55α holoenzymes during mitotic exit in cycling cells. Knock-down of PP2A B55α subunit leads to impaired and delayed dephosphorylation of SAMHD1 at T592 during mitotic exit. HeLa cells were either transfected with control siRNA or simultaneously with a single siRNA targeting PP2A B55α subunit. Transfected cells were then arrested in mitosis using nocodazole and mitotic exit was induced chemically by adding flavopiridol (FP) in the presence of MG-132. Cells were harvested at the indicated time points over a period of 18 min. Whole-cell lysates were analyzed by immunoblotting using antibodies specific to the indicated proteins. For quantification, the signal of phosphorylated SAMHD1 ( = SAMHD1 pT592; signal marked with asterisks, compared to SAMHD1 pT592 signal in non-synchronized cells (see Supplementary Fig. 5f)) was normalized to total SAMHD1 and level of SAMHD1 pT592 within 18 min compared between siCtr- and siB55α-transfected cells. Immunoblot data shown are representative of two independent experiments, while the quantification graphs represent the mean ± SD of both experiments

**Fig. 5** PP2A-B55α holoenzymes dephosphorylate SAMHD1 at T592 in vitro and in cells. **a–c** PP2A-B55α holoenzymes dephosphorylate SAMHD1 at pT592 in vitro. GFP-trapped active PP2A trimer complexes or GFP only were retrieved from HEK293T cells. After incubation with recombinant SAMHD1 for the indicated time points, samples were assessed by immunoblotting using antibodies specific to the indicated proteins. **a** Before incubation with recombinant SAMHD1, undiluted (1/1) and 1/3 diluted beads were either pre-incubated with buffer or 50 nM okadaic acid (OA) for 10 min at 30 °C. Data are representative of two independent experiments. **b** Purified $PP2A_D$ was added as an additional condition. Anti-PP2A C immunoblots allow to directly compare the amount of C subunit in both PP2A complexes (PP2A-B55α trimer versus $PP2A_D$). The lanes shown are derived from the same blot, but were not adjacently loaded. Data are representative of two independent experiments. **c** Anti-C immunoblotting indicates the amount of retrieved PP2A C subunit in these complexes. Data are representative of two independent experiments. **d, e** Inhibition of PP2A by OA (**d**) or siRNA-mediated silencing (**e**) increases SAMHD1 phosphorylation at T592 in cycling HeLa cells. **d** HeLa cells were treated with the phosphatase inhibitor OA (2 nM), harvested at different time points and analyzed by immunoblotting. **e** HeLa cells were transfected with control siRNA (lane 1), simultaneously with three different siRNAs targeting all subunits of the PP2A-B55α trimer (lane 3) or with single siRNAs targeting individual PP2A subunits (lane 4-6). 52 h post-transfection, cells were harvested and analyzed by immunoblotting. Data shown are representative of three (**d**) or two (**e**) independent experiments, respectively. **f, g** Inhibition of PP2A by OA (**f**) or by siRNA-mediated silencing of PP2A B55α subunit (**g**) increases SAMHD1 phosphorylation at T592 in MDMs. **f** MDMs were treated with the phosphatase inhibitor OA (2 nM), harvested at different time points and analyzed by immunoblotting. Data shown represent three donors analyzed. **g** MDMs were transfected with control siRNA or siRNA targeting B55α. 48 h post-transfection, cells were harvested and whole-cell lysates were analyzed by immunoblotting (**g**, left panel). For quantification, the signal of PP2A B55α subunit was normalized to GAPDH and compared to the control (**g**, left panel). In parallel, RNA samples were collected and *PPP2R2A (B55α)* mRNA levels determined by RT-qPCR (**g**, right panel). Data were normalized to the reference gene *RPL13A*. Fold change of *PPP2R2A (B55α)* mRNA level to siCtr was calculated based on three technical replicates; the graph shows the mean ± SD of the technical replicates (right panel). Data shown represent one of three donors analyzed

(reduction of SAMHD1 pT592 signal to $42.5 \pm 2.1\%$ for siCtr compared to $66 \pm 5.7\%$ for siB55α; $n = 2$) (Fig. 6). Therefore, we conclude that SAMHD1 is specifically targeted by PP2A-B55α holoenzymes in cycling cells during a short time window at the end of mitosis ($=$ mitotic exit). We propose that SAMHD1 is converted into an active restriction factor during mitotic exit, thereby providing an antiviral-active state during $G_1$ phase in cycling cells.

**Reduction in HIV-1 RT products in $G_1$ phase.** Our results demonstrate that PP2A-B55α holoenzymes are responsible for dephosphorylating SAMHD1 at T592 during mitotic exit in cycling cells (Fig. 6). Additionally, entering $G_1$ phase is accompanied by a notable and fast decrease in dATP levels (Fig. 1c and Supplementary Fig. 5c, d). In fact, we observed a significant decline for all four dNTPs upon $G_1$ entry compared to S ($= 2$ h post-release) and $G_2$/M phase ($= 7$ h post-release) (Supplementary Fig. 6a–c). Specifically, dCTP level are particularly low in $G_1$ phase ($4.22 \pm 0.33$ pmol/$10^6$ cells) compared to the low levels observed for the other dNTPs (dATP ($17.97 \pm 4.83$ pmol/$10^6$ cells), dGTP ($16.32 \pm 1.96$ pmol/$10^6$ cells) and dTTP ($25.6 \pm 5.03$ pmol/$10^6$ cells); all dNTPs: $n = 3$, Supplementary Fig. 6a). However, dNTP levels in $G_1$ phase are still higher when compared to the absolute levels in primary MDMs (Supplementary

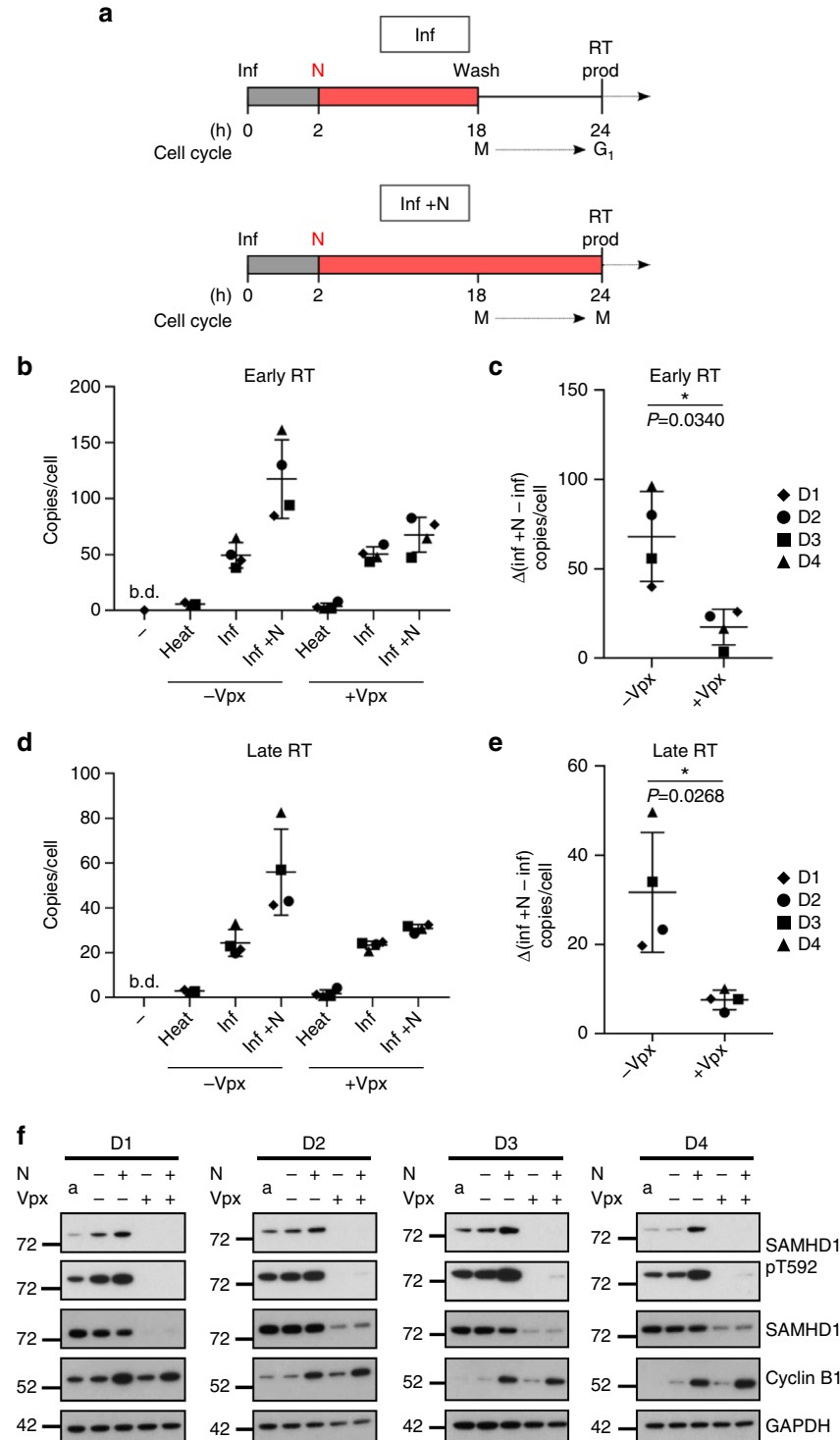

Fig. 7a, c) and resting CD4$^+$ T cells (Supplementary Fig. 7b, c). Moreover, when taking into account the differences in cell volumes of the various cell types, HeLa cells in G$_1$ phase contain ~21–35-fold and ~7–11-fold more dCTP than macrophages and resting CD4$^+$ T cells, respectively (Supplementary Fig. 7d), questioning whether the observed dNTP levels in G$_1$ phase may influence the critical threshold required for RT.

As a consequence, we were interested whether even in cycling cells dephosphorylated SAMHD1 would be able to reduce HIV-1 infection. Therefore, we synchronized HeLa cells using a double-thymidine block and infected at different time points post-release (Supplementary Fig. 6d). Notably, the points of infection were specifically chosen to ensure that onset of RT coincided with different cell cycle-phases and SAMHD1 phosphorylation states. Total DNA was harvested 4 h post-infection to prevent cells from progressing further in the cell cycle. During infection from S to G$_2$/M phase ("inf 1", 2–6 h post-release), SAMHD1 remains highly phosphorylated at T592 (Supplementary Fig. 6f) and cells are readily infected by HIV-1 (Supplementary Fig. 6e). In contrast, we detected less early RT products and a significant drop in late RT products (Supplementary Fig. 6e), when cells enter G$_1$ phase during the infection period ("inf 2"). The reduction in RT products is mirrored by a decrease in SAMHD1 phosphorylation at T592 ("inf 2", 7–11 h post-release; Supplementary Fig. 6f). Strikingly, infection with HIV-1 in the presence of Vpx (Supplementary Fig. 6g), which leads to degradation of SAMHD1 (Supplementary Fig. 6h), does not change the levels of RT products; the infection rates are comparable between G$_2$/M and G$_1$ phase (Supplementary Fig. 6g). In conclusion, reduction of RT products in cycling HeLa cells depends on the presence of SAMHD1 dephosphorylated at T592 (Supplementary Fig. 6i).

To validate that SAMHD1 could be antivirally active in the G$_1$ phase of primary target cells for HIV-1, we infected activated CD4$^+$ T cells with HIV-1 and subsequently arrested the cells in mitosis using nocodazole (Fig. 7a). Subsequently, nocodazole was either removed after 16 h, allowing CD4$^+$ T cells to exit mitosis/enter G$_1$ phase ("inf"), or nocodazole treatment was continued until 24 h post-infection ("inf + N"; Fig. 7a) to keep cells in mitosis. Indeed, continued nocodazole treatment leads to increased SAMHD1 phosphorylation at T592 in all donors analyzed compared to non-synchronized, activated CD4$^+$ T cells (Fig. 7f, comparing " + N" to condition "a"). In contrast, SAMHD1 phosphorylation at T592 is reduced after nocodazole wash-out, as CD4$^+$ T cells were able to enter G$_1$ phase, which is also indicated by decreasing cyclin B1-levels (Fig. 7f, "-N"). Interestingly, we observed higher early and

late HIV-1 RT products upon HIV-1 infection in absence of Vpx (-Vpx) in mitotic cells, which displayed high SAMHD1 pT592-levels ("inf + N"), than compared to CD4$^+$ T cells that entered G$_1$ phase characterized by a reduced pT592 signal ("inf") (Fig. 7b, d, "-Vpx"). Again, as seen in HeLa cells, infection with HIV-1 in the presence of Vpx leads to SAMHD1 degradation (Fig. 7f) and a significantly less pronounced difference in HIV-1 products (Fig. 7b, d, " + Vpx"; Fig. 7c, e). Therefore, we conclude that the observed reduction of HIV-1 RT products upon G$_1$ entry in cycling CD4$^+$ T cells is depending on the presence of dephosphorylated, active SAMHD1.

Since SAMHD1 dephosphorylation at T592 by PP2A-B55α holoenzymes reduces HIV-1 infection efficiency, we tested whether HIV accessory proteins actively alter B55α subunit expression. No significant modulation of endogenous B55α levels could be detected (Supplementary Fig. 8), whereas B56δ subunit expression was counteracted by Vif as described recently[41].

Taken together, our results demonstrate that PP2A is important for controlling SAMHD1 dephosphorylation at T592, in both cycling and non-cycling cells, and in consequence for its restrictive activity against HIV-1.

## Discussion

SAMHD1-mediated restriction of HIV-1 replication in non-dividing cells is part of the antiretroviral defense program. This restriction is strictly regulated by dephosphorylation of T592 in SAMHD1[21–23]. The dephosphorylation at this specific residue might also impact SAMHD1 functions, such as the ability to degrade RNA[6], tetramerize and hydrolyze dNTPs[25–27], although this has been controversially discussed[8,22,23,28,42,43]. Phospho-mimetic mutants of SAMHD1 are able to deplete the cellular levels of dNTPs[22,23,42], suggesting that phosphorylation of this specific residue has no influence on the dNTPase function of SAMHD1. Nevertheless, these mutants abrogate viral inhibition, furthermore supporting the importance of post-translational regulation of SAMHD1 at T592 in HIV-1[21–23,42] and also HBV replication[15]. Moreover, phospho-regulation at T592 impacts replication fork progression, degradation of nascent DNA at arrested forks and chronic inflammation[29].

SAMHD1 dephosphorylation at T592 is regulated during cell cycle-progression in proliferating cells. We show that in synchronized HeLa cells, phosphorylation at T592 starts building up at the G$_1$/S border and remains constant until mitosis, which is in line with the requirements for phosphorylated SAMHD1 as a

---

**Fig. 7** SAMHD1 dephosphorylation at T592 upon G$_1$ entry correlates with a decrease in HIV-1 RT products in activated primary CD4$^+$ T cells. **a** Schematic of infection experiment in activated primary CD4$^+$ T cells. After activation for 5 days by PHA-P/IL-2, primary CD4$^+$ T cells were infected with VSV-G-pseudotyped HIV-1 reporter virus (-/+ Vpx) for 2 h. After removal of the virus, cells were arrested in mitosis by nocodazole (N) treatment. Subsequently, CD4$^+$ T cells were washed twice 18 h post-infection (p.i.) and, as a result, upon removal of nocodazole cells could progress to G$_1$ phase (condition "inf"), whereas upon continuation of nocodazole treatment, cells remain in mitosis (condition "inf + N"). Total DNA for RT product measurements was harvested 24 h p.i; MOI 7.5). Importantly, the time of removal or continuation of nocodazole was chosen in such a way that sufficient DNA copies at 16 h p.i. could be detected and coincided with specific cell cycle-phases. **b, d** Increased SAMHD1 phosphorylation at T592 after mitotic arrest correlates with higher HIV-1 RT copy numbers. Primary CD4$^+$ T cells were infected with VSV-G-pseudotyped HIV-1 luciferase reporter virus (-/+ Vpx) or heat-inactivated virus and subsequently treated with nocodazole as described in Fig. 7a. 24 h p.i., total DNA was collected and the amount of early (**b**) and late (**d**) RT products determined by qPCR. Each sample was measured in technical triplicates. Data shown represent the mean ± SD of four donors analyzed (each depicted by a specific symbol). **c, e** Increase in early (**c**) and late (**e**) HIV-1 RT products with (continued) nocodazole treatment depends on the absence of Vpx, correlating with phosphorylated SAMHD1 (Fig. 7f). Differences in RT products ( = Δ(inf + N – inf)) observed in Fig. 7b and d, respectively, after infection with VSV-G-pseudotyped HIV-1 reporter virus in the absence and presence of Vpx were calculated (−/ + Vpx). Statistical significance was determined using a paired, two-tailed Student's t-test (ns: $p \geq 0.05$; *: $p < 0.05$; **: $p < 0.01$; ***: $p < 0.001$). Data shown represent the mean difference ± SD of four donors analyzed (each depicted by a specific symbol). **f** Validation of SAMHD1 pT592-status and degradation of SAMHD1 upon Vpx delivery in activated primary CD4$^+$ T cells (related to Fig. 7b-e). For immunoblotting, primary CD4$^+$ T cells were harvested after activation for 5 days ( = a), after release from nocodazole arrest ( = -N; 24 h p.i.) and with continued nocodazole treatment ( = + N; 24 h p.i.). Whole-cell lysates were analyzed using antibodies specific to the indicated proteins. Data shown represent four donors analyzed

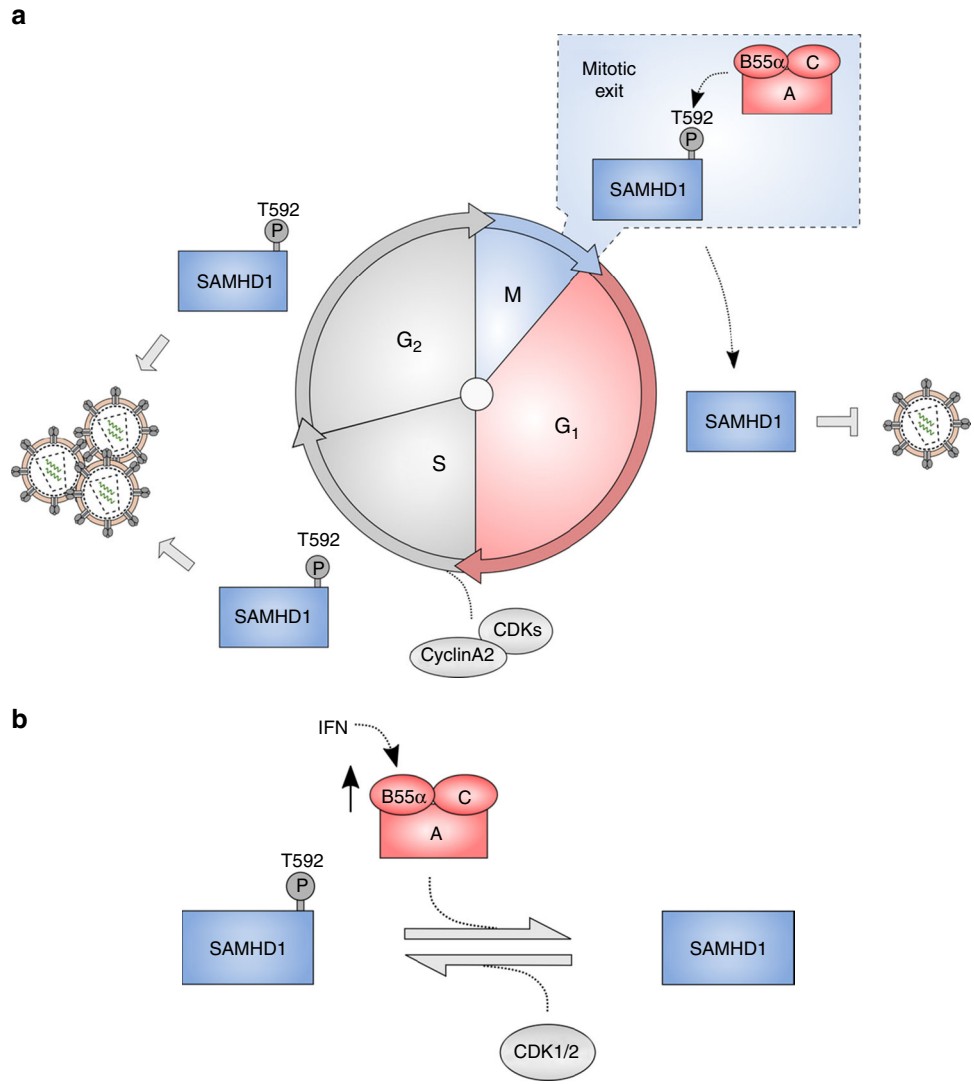

**Fig. 8** Regulation of SAMHD1 T592 phosphorylation in cycling and differentiated cells. **a** Regulation of SAMHD1 T592 phosphorylation in cycling cells. Initially, phosphorylation at T592 in SAMHD1 is introduced by CDK2/cyclin A2 in S phase and maintained by CDK1/cyclin A2 until mitosis. SAMHD1 T592 phosphorylation is rapidly lost upon $G_1$ entry, more specifically during mitotic exit, through dephosphorylation by PP2A-B55α holoenzymes. Dephosphorylated SAMHD1 can reduce or delay viral cDNA synthesis upon HIV-infection in the $G_1$ stage—even in cycling cells. **b** Regulation of SAMHD1 T592 phosphorylation in differentiated MDMs. Control of T592 phosphorylation level in SAMHD1 is exerted through differential CDK1/2 activities, potentially depending on the macrophage type. Furthermore, PP2A-B55α holoenzymes are involved in balancing SAMHD1 T592 phosphorylation in differentiated MDMs. IFN leads to upregulation of PP2A B55α subunit, an additional layer of regulation to control SAMHD1 antiviral activity

promoter for fork resection in S phase[29]. Upon re-entry of interphase, SAMHD1 is rapidly dephosphorylated at T592 (Fig. 1 and Supplementary Fig. 1), which is in concordance with a previous report by Yan et al. detecting a loss of phosphorylated T592 in $G_1$-sorted cell populations of proliferating cells[27]. Moreover, we show a concomitant drop of all four dNTPs upon entering $G_1$ phase (Fig. 1c; Supplementary Fig. 6a), reinforcing that dNTP levels are tightly regulated at the transition from M to $G_1$. As dephosphorylation of SAMHD1 at T592 happens concurrently, this suggests either a coincidental or causative relationship between dephosphorylation and dNTPase activity. Tight cell cycle-regulation is in line with the cellular requirements for balancing dNTP levels during cell cycle-progression[44]. It could be hypothesized that SAMHD1 as the counterpart to the ribonucleotide reductase (RNR), which synthesizes dNTPs de novo[45], becomes an active dNTPase in the $G_1$ phase (Fig. 1c), whereas the R2 subunit of RNR is absent in $G_0/G_1$ upon its degradation in mitosis. Furthermore, we extended this observation by analyzing

the exact temporal transition between M and $G_1$ phase. We chemically induced mitotic exit events in synchronized cells by inactivation of CDKs in absence of proteasome-mediated degradation. SAMHD1 dephosphorylation at T592 happens during mitotic exit (Supplementary Fig. 5b)—a complex, irreversible and unidirectional process that starts with central spindle assembly and chromosome segregation. This process is highly regulated (i) through degradation of mitotic factors and (ii) by phosphatases that remove phosphorylations from mitotic substrates[32]. Intriguingly, concurrently with SAMHD1 dephosphorylation at T592, dATP levels dropped rapidly within 18 min upon forced exit out of mitosis (Supplementary Fig. 5c, d). As this assay was performed in the presence of a proteasomal inhibitor, the degradation of the R2 subunit of the RNR or possibly other not yet discovered factors responsible for regulating cellular dNTP pools might not be held accountable for this rapid decline in dNTPs. But at this point, we cannot rule out that not yet identified metabolic enzymes or (co)-factors regulating the decline in dNTP pools might get activated

during mitotic exit. Since SAMHD1 expression levels did not change within different cell cycle-phases (Fig. 1), consistent with reported observations in sorted, unsynchronized human primary CD4$^+$ T cells and monocytic cell lines[27,46], we suggest that SAMHD1 function is not regulated by fluctuating expression levels (as it was proposed by[45,47]). Instead, we propose that SAMHD1's diverse functions are regulated throughout the cell cycle by post-translational modifications. Further studies are needed to determine how T592 dephosphorylation mechanistically modulates SAMHD1's enzymatic activities and functions in vivo, specifically the antiviral mechanism. We cannot rule out that exit from mitosis itself or the dephosphorylation event at T592 might trigger cell cycle-specific protein interactions, possibly in interplay with additional post-translational modifications[46], or nucleic acid binding that would influence the enzymatic functions of SAMHD1 during G$_1$ phase.

Intriguingly, our studies revealed the responsible phosphatase PP2A that renders SAMHD1 restrictive against HIV-1. As PP2A substrate specificity is defined by the regulatory B-type subunits[48], we revealed that PP2A holoenzymes containing the specific subunit B55α dephosphorylate SAMHD1 at T592 during mitotic exit (Fig. 6). In PP2A-B55α-depleted cells, the removal of phosphorylation at CDK substrates and SAMHD1 phosphorylation at T592 was delayed in comparison to control cells (Fig. 6 and Supplementary Fig. 5). The results support the conclusion that T592 in SAMHD1 is a mitotic substrate of PP2A-B55α trimers, and are in line with the fact that this particular holoenzyme represents a key mitotic exit phosphatase in mammalian cells[39]. Importantly, the discovery of PP2A-B55α as the responsible phosphatase acting on T592 in SAMHD1 during mitotic exit is in line with current knowledge on the timely coordinated and conserved program that occurs in mammalian cells during mitotic exit[49] (see Supplementary Note 1).

We identified key basic residues flanking T592, a unique recognition site shared only by B55 substrates, responsible for interaction between SAMHD1 and the holoenzyme PP2A-B55α (Fig. 4e, f). Interestingly, mitotic exit substrates are recognized in a temporal order encoded in this site. The phosphatase decodes this signal through a complementary negatively charged surface on the regulatory subunit B55[37]. Additionally, it was shown that the phospho-amino acid threonine over serine determines the turn-over rate during mitotic exit[37,50], suggesting that threonine at residue 592 determines human SAMHD1 as an ideal strong B55 substrate. An interesting side aspect comparing human to murine SAMHD1 (Fig. 4e, f) suggests that murine isoform 1[51,52] may (i) be a low affinity substrate of PP2A-B55 and (ii) may display slower dephosphorylation kinetics compared to human SAMHD1 as it contains much less basic residues. Of note, human and murine B55α subunit display 100% sequence identity. This would argue for a different regulation of murine compared to human SAMHD1. The identified key residues critical for binding of human SAMHD1 to PP2A are situated in the C-terminus, a platform for interaction with cellular and viral partners[53–55], and are highly accessible based on the structural model (see Supplementary Note 1).

Importantly, as soon as proliferating cells enter the G$_1$ cell cycle-stage, HIV-1 infection resulted in significant reduction of RT products depending on the presence of SAMHD1 dephosphorylated at T592 (Fig. 7 and Supplementary Fig. 6d-i), while concomitantly dNTP levels are low (Fig. 1c; Supplementary Fig. 5c; Supplementary Fig. 6a-c). We propose the following model (Fig. 8a): even in proliferating cells, the G$_1$ stage of the cell cycle represents a short window where SAMHD1 could be antivirally active and delay RT kinetics. This could explain previous observations on restrictive SAMHD1 in proliferating cells[20,56]. Our results suggest that SAMHD1 is dephosphorylated by PP2A-

B55α trimers in cycling cells every time cells exit mitosis and enter interphase. Active SAMHD1 in G$_1$ then could delay RT of HIV-1. Whether the reduced cellular pool of dNTPs is responsible for HIV-1 inhibition in G$_1$ phase, the proposed controversial nuclease activity[4,5] or nucleic acid-binding properties[2,3] triggered upon mitotic exit, will need to be determined. In fact, the calculated dNTP concentrations in cells (mainly) refractory to HIV-1 infection, macrophages and resting CD4$^+$ T cells, are lower than the measured dNTP levels in the G$_1$ phase in HeLa cells (Supplementary Fig. 7d). When taking into account the range of $K_m$ and $K_d$ values of HIV-1 RT[57], this suggests that the lowest measured level in G$_1$ phase that would be rate-limiting (1.09–1.8 μM dCTP) lies still above the general $K_m$ values of HIV-1 RT and might not significantly affect the replication kinetics. Still, the limited substrate concentrations could influence dNTP binding affinity. Particularly, highly structured RNA templates such as retroviral RNA are known to interrupt processive DNA synthesis resulting in stalling of RT[58]. This mechanism is influenced by restricted substrate availability[59] and could contribute to the observed reduction of RT in G$_1$ phase. On the other hand, it is likely that additional mechanism of restriction by SAMHD1 play a role that are triggered upon mitotic exit and correlate with dephosphorylated SAMHD1 at T592. The latter hypothesis would agree with Welbourn et al.[42] who provide evidence for a dNTPase-independent function of SAMHD1. This function particularly contributes to the lentiviral restriction observed in HeLa cells where dNTP levels were artificially lowered by inhibiting the RNR[42], paralleling our experiment in synchronized cells providing a state of low dNTP levels in G$_1$ phase. In contrast, SAMHD1 is phosphorylated from S to M phase (Fig. 1a) and rendered inactive against HIV-1. We speculate that HIV-1-infected cells are reprogrammed to avoid progression to the end of mitosis, where PP2A-B55α holoenzymes would be active and could remove CDK-mediated phosphorylations from targets like SAMHD1. Thereby, HIV-1 actively evades dephosphorylated, restrictive SAMHD1 present in G$_1$ phase in cycling cells. As HIV-1 capsid uncoating is triggered only after first strand transfer of RT[60] and as SAMHD1 restriction is a reversible process[61], it might be likely that HIV-1 capsids are able to "wait around" for the S phase to continue the infection process.

Intriguingly, HIV-1 actively manipulates cell cycle-progression by inducing a G$_2$ arrest mediated by the accessory protein Vpr[62]. First indications suggested an involvement of PP2A as (i) okadaic acid and (ii) regulatory and catalytic subunits of PP2A were shown to influence the Vpr-induced G$_2$ arrest[63,64]. Interestingly, the HIV-1 accessory protein Vif has been reported to affect specific PP2A holoenzymes by directing the B56 regulatory subunits to CUL5-dependent proteasomal degradation[41]. Indeed, our results confirm Vif-dependent influence on B56 subunit expression (Supplementary Fig. 8); however, our proofs indicate no striking influence on PP2A-B55α subunit expression by any of the accessory proteins (Supplementary Fig. 8). Intriguingly, in fission yeast, it was demonstrated that the events during mitotic exit are driven by a coordinated and sequential activation of various phosphatases, the so-called phosphatase relay, where PP1 activation is required for the reactivation of both PP2A-B55 and PP2A-B56 and, in turn, full activation of the phosphatase relay[65]. Given the conservation of PP1 docking sites in both subunits across species[66], proper control of mitotic exit events may indeed be dependent on a phosphatase relay. This would suggest that antagonism of PP2A-B56 by HIV-1 Vif could possibly influence PP2A-B55α action during mitotic exit and possibly represent another level of active manipulation of the cell cycle besides the mentioned G$_2$/M arrest by Vpr, in order to avoid antivirally active SAMHD1 in the G$_1$ stage.

Moreover, we provide first evidence that PP2A is involved in controlling the level of SAMHD1 T592 phosphorylation in differentiated MDMs (Fig. 5f, g). We propose the following model (Fig. 8b): MDMs contain active kinases that are constantly kept in check by the action of PP2A to hold a balance of dephosphorylated targets, such as SAMHD1. This is supported by the fact that silencing of various CDKs in MDMs has been reported to reduce phosphorylation of T592[30]. Intriguingly, we could show that the PP2A B55α subunit was specifically upregulated at mRNA and protein level by IFNs in MDMs (Supplementary Fig. 4a, b; see Supplementary Note 1), indicating that PP2A-B55α holoenzymes are preferentially involved in dephosphorylating SAMHD1 after IFN stimulation in MDMs. This would explain the reported observation that a phosphatase activity might be inducible by IFNs in MDMs[21], which reflects another layer of control over SAMHD1's restrictive capacity against HIV-1.

Taken together, our findings suggest PP2A-B55α as the key regulator to dephosphorylate SAMHD1 at T592 in MDMs and during mitotic exit in proliferating cells, thereby converting SAMHD1 into an active HIV-1 restriction factor. Understanding the activation process of SAMHD1 will facilitate the development of new therapeutic strategies to control virus replication and to regulate the innate immune system.

## Methods

**Ethics statement.** Human buffy coats of anonymous blood donors were purchased from the German Red Cross Blood Donor Service Baden-Württemberg Hessen.

**Cell lines and human primary cells.** Human HeLa 'Kyoto'[39] (resource identification: RRID:CVCL_1922) and HEK 293 T (ATCC No.: CRL-3216) or 293 T/17 (ATCC No.: CRL-11268) cells were grown in DMEM (Lonza) supplemented with 10% (v/v) fetal bovine serum (FBS; Sigma-Aldrich) and 2 mM L-glutamine (Biochrom) at 37 °C and 5% $CO_2$. Human monocytic THP-1 (ATCC No.: TIB-202) cells were grown in RPMI-1640 (Biowest) supplemented with 10% (v/v) FBS and 2 mM L-glutamine at 37 °C and 5% $CO_2$. All the cell lines were regularly tested for the absence of mycoplasma using the MycoAlert Mycoplasma Detection Kit (Lonza). Authentication of cell lines was confirmed by examining genetic characteristics through PCR-single-locus-technology (Eurofins Medigenomix Forensik).

Human peripheral blood mononuclear cells (PBMCs) were isolated by density gradient centrifugation (980g, 30 min, RT) using Histopaque-1077 (Sigma-Aldrich). PBMCs were diluted in PBS, centrifuged (550g, 10 min, RT) and cell pellets were resuspended in 10 mL 0.87% ammonium chloride for lysis of erythrocytes (10 min, 37 °C). PBMCs were washed twice with PBS and the amount of isolated cells was determined. Monocytes were purified from PBMCs using CD14 MicroBeads (Miltenyi Biotec) according to the manufacturer's instructions. CD4+ T cells were purified from PBMCs using CD4 MicroBeads (Miltenyi Biotec) according to the manufacturer's instructions. Separation from unlabeled cells was carried out using an autoMACS Pro Separator (Miltenyi Biotec).

For the generation of monocyte-derived macrophages (MDMs), isolated monocytes were cultured in RPMI-1640 supplemented with 10% FBS, 2 mM L-glutamine, 10 mM HEPES, 1 mM sodium pyruvate and 100 U/mL granulocyte-macrophage colony-stimulating factor (GM-CSF; PeproTech). After 72 h, fresh medium containing the respective cytokines was added. On day 5, all cells were harvested and the amount of differentiated cells determined using the Cellometer Auto T4 (Nexcelom Bioscience).

For activation of CD4+ T cells, isolated cells were cultured at a density of 3 × 10⁶ cells/mL in RPMI-1640 supplemented with 10% FBS, 2 mM L-glutamine, 1% Pen-Strep and 25 mM HEPES, 500 U/mL Proleukin S (Novartis) and 5 μg/mL phytohemagglutinin (PHA-P; Sigma-Aldrich) for 5 days, including renewal of the culture medium/supplements 72 h after start of activation.

**IFN treatment and RT-qPCRs.** For IFN stimulation, 7.5 × 10⁵ cells/12-well (RNA samples for quantitative real-time reverse transcription PCR (RT-qPCR)) or > 5 × 10⁵ cells/12-well (whole-cell lysates for immunoblotting) were seeded. MDMs were left untreated or treated with 1000 U/mL IFNα2a (PeproTech), IFNβ1a (PBL Interferon Source) or IFNγ (PeproTech) for 8/24 h (RNA samples) or 24 h (whole-cell lysates) at 37 °C. As a control for IFN induction, MDMs were infected with Sendai virus (final dilution = 1:200).

Total RNA from MDMs was isolated using the RNeasy Plus Mini Kit (QIAGEN) or NucleoSpin RNA Kit (Macherey-Nagel). Expression of mRNAs was determined in a 384-well format using QuantiTect SYBR Green RT-PCR Kit (QIAGEN) on an ABI7900 cycler (Applied Biosystems). Isoform-specific primers are listed in the Supplementary Methods.

**HIV-1 infection and viral DNA quantitation by qPCR.** 3 × 10⁴ HeLa cells/12-well were seeded and synchronized using a double-thymidine block. After the 2nd release, cells were counted at each time point to ensure the use of equal amounts of virus. Subsequently, cells were infected with a VSV-G-pseudotyped, full-length HIV-1 reporter viruses mutated in p6 allowing for production of virions that package SIVmac239 Vpx[67] and lacking Vpr (pNL4.3 E⁻R⁻ luc3 chp6_pcDNA, MOI 3; pNL4.3 E⁻R⁻ luc3 chp6_Vpx, MOI 1.5). Prior to infection, HIV-1 virus stocks were incubated with 10 U/mL DNaseI (NEB) (30 min, 37 °C) to reduce contamination by plasmid DNA in subsequent qPCR assays. For control purposes, the used virus was heat-inactivated (65 °C, 20 min). Cells were spin occulated (30 min, 245g, 32 °C) and the virus removed 1 h post-infection. In order to monitor cell cycle-phases/SAMHD1 phosphorylation states, samples for immunoblotting were harvested at the time of infection and DNA harvest.

2 × 10⁵ activated CD4+ T cells/96-well were seeded and infected with a VSV-G-pseudotyped, full-length HIV-1 reporter viruses -/+Vpx lacking Vpr (pNL4.3 E⁻R⁻ luc chp6_pcDNA and pNL4.3 E⁻R⁻ luc chp6_Vpx, MOI 7.5 each). Prior to infection, HIV-1 virus stocks were incubated with 10 U/mL DNaseI (NEB) (30 min, 37 °C) to reduce contamination by plasmid DNA in subsequent qPCR assays. For control purposes, the used virus was heat-inactivated (95 °C, 10 min). Cells were spin occulated (1.5 h, 800g, 32 °C) and the virus removed 2 h post-infection. With removal of virus, cells were arrested using 100 ng/mL nocodazole (Sigma-Aldrich) for 16 h ( = inf) or 24 h ( = inf + N) at 37 °C. In order to allow cell cycle-progression from mitosis into G₁ phase, arrested CD4+ T cells were washed twice with warm medium to remove nocodazole (after 16 h) and again incubated for 6 h at 37 °C ( = inf). In order to monitor cell cycle-phases/SAMHD1 phosphorylation states, samples for immunoblotting were harvested prior to infection and 24 h post-infection ( = DNA harvest).

For both cell types, virus was titrated in a time-course experiment to detect the onset of RT products upon 3 h after infection in HeLa cells and upon 16 h after infection in activated CD4 + T cells, respectively.

Total DNA was isolated 4 h (HeLa) or 24 h post-infection (CD4+ T cells) post-infection using the DNeasy Blood & Tissue Kit (QIAGEN). DNA was subjected to qPCR to specifically quantify HIV-1 early and late RT products. To normalize the amount of input DNA, the copy number of the cellular gene porpholilinogen deaminase (PBGD) was quantified using a standard curve generated in the respective experiment.

**Plasmids.** A full list of plasmids used in this study and detailed protocols describing respective cloning procedures can be found in the Supplementary Methods.

**Double-thymidine block.** For synchronization, 0.5 × 10⁶ HeLa cells were seeded per 10-cm-cell-culture dish. 24 h after seeding, the medium was replaced with DMEM + 2 mM thymidine and cells were incubated for 16 h at 37 °C ( = 1st block). Cells were washed and released for 8 h at 37 °C into DMEM + 25 μM 2'-deoxycytidine ( = 1st release). The procedure was repeated ( = 2nd block/2nd release) and cells were harvested at different time points post-release. The respective samples were split for immunoblot analysis and determination of cell cycle-phases by propidium iodide (PI) staining/flow cytometry.

**Chemically induced mitotic exit assay.** For the chemically induced mitotic exit assay[39], 4–5 × 10⁵ HeLa cells were seeded in 5 mL DMEM per 6-cm-cell-culture dish. 17 h before chemical induction of mitotic exit ( = day 2), HeLa cells were arrested using nocodazole (100 ng/mL). On day 3, arrested cells were incubated for 30 min in DMEM containing 30 μM MG-132 and collected by mitotic shake-off. Cells were washed, resuspended in PBS (containing 30 μM MG-132) and divided into aliquots. Subsequently, cell aliquots were incubated at 37 °C and forced to exit mitosis by addition of flavopiridol (final concentration: 20 μM). Single cell aliquots were lysed every 3 min (over a total time period of 18 min) using radio-immunoprecipitation assay (RIPA) buffer (supplemented with protease and phosphatase inhibitors) and directly frozen at -20 °C.

**RNA interference (RNAi).** For immunoblot analysis, 2 × 10⁴ HeLa cells were seeded in 500 μL DMEM per 24-well and incubated overnight at 37 °C. HeLa cells were transfected with isoform-specific small interfering RNAs (siRNAs) for different PP2A subunits or scrambled controls (see Supplementary Methods) using Lipofectamine RNAiMAX Reagent (Invitrogen) according to the manufacturer's instructions (siRNA amount/well: 5 pmol). 52 h post-transfection, cells were harvested for immunoblot analysis.

For the chemically induced mitotic exit assay, 5 × 10⁵ HeLa cells were seeded in 5 mL DMEM per 6-cm-cell-culture dish and incubated overnight at 37 °C. 52 h before chemical induction of mitotic exit, HeLa cells were transfected as described above (siRNA amount/dish: 50 pmol).

For siRNA-transfection of MDMs, 3 × 10⁵ MDMs/24-well were seeded in 500 μL RPMI-1640 (supplemented with 10% FBS, 2 mM L-glutamine, 10 mM HEPES, 1 mM sodium pyruvate) and allowed to re-attach for 2 h at 37 °C. MDMs were transfected with siRNAs specific to PP2A B55α subunit, PP2A Cα subunit or a scrambled control using Stemfect RNA Transfection Kit (Stemgent; siRNA amount/well: 10 pmol). After 24 h, the medium was changed and MDMs were

again transfected with siRNAs as described above. 48 h or up to 5 days post-transfection, MDMs were harvested for immunoblot and RT-qPCR analysis.

**Mass spectrometry to identify SAMHD1-interacting proteins**. Two independent mass spectrometry (MS) approaches were performed (Fig. 2a: GFP-traps subjected to on-bead trypsin digestion and Supplementary Fig. 2: in-gel digestion of tandem-affinity purified CBP-SBP-SAMHD1, see Supplementary Methods for the latter approach). For GFP-trapping, $3 \times 10^6$ HEK293T cells/10-cm-cell-culture dish were seeded. After 24 h, HEK293T cells were transfected with 6 µg (pEGFP-C1-SAMHD1 or pEGFP-C1) of total plasmid DNA per dish using 18 mM PEI reagent. 48 h post-transfection, cells were harvested, pooled and lysed in 1 mL NET lysis buffer/sample.

Anti-GFP trapping of the same amount of lysates from GFP or GFP-SAMHD1 overexpressing cells was executed; notably, GFP-trapped complexes were washed stringently for four times using NENT300 (20 mM Tris [pH 7.4], 300 mM NaCl, 1 mM EDTA [pH 7.4], 0.1% NP40, 25% glycerol). GFP-traps were subjected to on-bead trypsin digestion and the resulting peptide mixture was analyzed by LC-MS/MS on a nano-LC hybrid quadrupole-orbitrap mass spectrometer (Q Exactive, Thermo Fisher Scientific). Relative quantification of proteins in different conditions was executed with Progenesis software (Nonlinear Dynamics) incorporating protein identifications obtained by MASCOT (Matrix Science) search engine using Swiss-Prot (*Homo sapiens*, 20,202 entries) as a database. Only peptides with 95% peptide identification probability, resulting in a 1% peptide FDR (Scaffold), were taken into account during the analysis. As a result, a list of proteins present in the GFP-SAMHD1 trap and not present in the GFP trap could be established (280 hits). CDK1 and the PP2A Aα ( = 65 kDa) subunit were present in this list.

**Co-immunoprecipitation (CoIP)**. With transfected nFLAG-SAMHD1 in HEK293T cells. For CoIPs, $3 \times 10^6$ HEK293T cells were seeded per 10-cm-cell-culture dish. After 24 h, HEK293T cells were transfected with 6 µg (pcDNA3.1( + )-nFLAG-SAMHD1 only) or 11 µg (5.5 µg pcDNA3.1( + )-nFLAG-SAMHD1 + 5.5 µg PP2A B-type subunits in pEGFP ( = GFP-tagged)) of total plasmid DNA per dish using 18 mM PEI reagent. 48 h post-transfection, cells were harvested and lysed in 200 µL NET lysis buffer (containing protease and phosphatase inhibitors) per dish for 30 min on ice. Lysates were pre-cleared and subsequently incubated with 25 µL ANTI-FLAG M2 Affinity Gel (Sigma-Aldrich) for 1 h at 4 °C. After washing, bound immune complexes released in 25 µL LDS Sample Buffer through boiling (95 °C, 5 min).

With transfected GST-tagged PP2A B subunits in HEK293T cells. For CoIPs, $3 \times 10^6$ HEK293T cells were seeded per 10-cm-cell-culture dish. After 24 h, HEK293T cells were transfected with 11 µg of total plasmid DNA (3.5 µg pcDNA3.1( + )-nFLAG-SAMHD1 + 7.5 µg PP2A B-type subunits in pGMEX-T1 ( = GST-tagged)) per dish using 18 mM PEI reagent. 48 h post-transfection, cells were harvested and lysed in 200 µL NET lysis buffer (containing protease and phosphatase inhibitor) per dish for 30 min on ice. Lysates were incubated with 25 µL GST-Trap_M beads (Chromotek; Nano-Traps consisting of a second type of antibody of the *Camelidae* species, called heavy chain antibodies (hcAbs), that bind their antigen through a single variable domain (VHH) devoid of light chains (so-called nanobody), coupled to magnetic microparticles) for 1.5 h at 4 °C. Subsequently, beads were washed and bound immune complexes released in 25 µL LDS Sample Buffer through boiling (95 °C, 5 min).

**In vitro-dephosphorylation assays and targeted LC-MS/MS**. De novo purified $PP2A_D$ and PP1 were obtained from a commercial source (Millipore). For comparative MS analysis, 1 µg of recombinant SAMHD1 was incubated in enzyme dilution buffer (0.15 M NaCl, 20 mM MOPS, pH 7.5, 60 mM 2-mercaptoethanol, 0.1 mM MnCl₂, 1 mM MgCl₂, 1 mM EGTA, 10% glycerol and 0.1 mg/mL serum albumin) with 0.032 units/µL PP2A, 0.032 units/µL PP1 or buffer only, for 1 h at 30 °C, in a reaction volume of 10 µL. 1 µL of complete protease inhibitor mix (Roche) was also included. Reactions were stopped by adding 5 × SDS-PAGE sample buffer and 10 min of boiling. Following SDS-PAGE of the samples, the 1D-gel piece containing SAMHD1 was subjected to in-gel trypsin digestion followed by C18 peptide cleanup (ZipTip, Millipore). The resulting peptide mixture was subjected to targeted LC-MS/MS on a nano-LC hybrid quadrupole-orbitrap mass spectrometer (Q Exactive, Thermo Fisher Scientific). Data analysis was executed with Pinpoint™ 1.4.0 (Thermo Fisher Scientific) software. Normalization of data in the different conditions was done based on a non-phosphorylatable peptide of SAMHD1 (GGFEEPVLLK) (Supplementary Table 1). Calculation of phosphorylation stoichiometry was executed as described in Schreurs et al.[68], and based on a method described by Olsen et al.[69]. For immunoblot analysis with anti-SAMHD1 pT592 antibodies, the amount of SAMHD1 was reduced to 0.1 µg per dephosphorylation reaction. For in vitro-dephosphorylation with PP2A trimers, the relevant GFP-tagged B-type subunits (B55α, B56α, B56β, PR72) were first expressed in HEK293T cells, and the respective catalytically competent PP2A trimers harboring a specific B-type subunit were retrieved from the transfected cells by GFP-trapping, as previously described[70]. Whenever OA (Calbiochem) was used in the assays, the phosphatases were first pre-incubated with 50 nM OA for 10 min at 30 °C, before their addition to the substrate. All dephosphorylation reactions were stopped by adding 5 × SDS-PAGE sample buffer and boiling.

Uncropped scans of immunoblots are supplied in the Supplementary Information.

**Statistical analysis**. Statistical analysis was performed using the Graph Pad Prism software (Version 7.03). Data are represented as mean ± standard deviation (SD). To assess statistical significance, a paired/unpaired, two-tailed Student's *t*-test was applied (ns: $p \geq 0.05$; *: $p < 0.05$; **: $p < 0.01$; ***: $p < 0.001$) or one-way ANOVA with multiple comparisons according to Dunnett/Sidak (ns: $p \geq 0.05$; *: $p < 0.05$; **: $p < 0.01$; ***: $p < 0.001$; P values were adjusted to account for multiple comparisons).

**Data availability**. All data are available from the corresponding author upon request.

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

## Acknowledgements

The authors thank Anna-Maria Engin, Nicole Gorgs, and Michaela Neuenkirch for technical assistance as well as Maximilian Rieß, Manja Burggraf, and Andreas Sommer for experimental support/critical discussion. Additionally, The authors thank Oliver T. Keppler (LMU Munich, Germany; for anti-SAMHD1 pT592 antibody), Sumit K. Chanda (Sanford Burnham Prebys Medical Discovery Institute, USA; for siRNAs), Nevan J. Krogan (University of California San Francisco (UCSF), USA; for plasmids encoding HIV-1 accessory proteins), Nathaniel R. Landau (NYU School of Medicine, USA; for pNL4.3 plasmids), Georg Kochs (University of Freiburg, Germany; for Sendai virus), Daniel Gerlich (University of Vienna, Austria; for HeLa 'Kyoto' cells), and Stephen Dilworth (Middlesex University London, UK; for monoclonal PP2A C antibodies) for reagents. This work is supported by Deutsche Forschungsgemeinschaft (DFG): CRC1292 project TP04 and SPP1923 project KO 4573/1-1 to R.K., BMBF 01KI1307A and German Center for Infection Research (DZIF) HZI2010Z10, DZIF TTU 01.802 to R.K. V.J. and R. D. were supported by the IAP Program of the Belgian federal government (P7/13) and the KU Leuven Research Council (OT/13/094). A.B.N. and F.D.G. were supported by an NIH R01 GM123540 grant to F.D.G. B.K., B.M. and C.S. were supported by R01 GM104198 and R01 AI136581 to B.K.

## Author contributions

R.K., V.J., R.D., and K.S. designed the research. K.S., N.V.F., R.D., J.S., E.S., C.T., H.S., A. R., and V.J. performed the experiments. C.S., B.M., and B.K. provided dNTP

measurements. A.B.N. and F.D.G. provided critical reagents and material. R.K., V.J., and K.S. interpreted the data and wrote the manuscript.

## Additional information

**Competing interests:** The authors declare no competing interests.

