## [Peer Review File · Nature Communications]

Reviewers' comments:

Reviewer #1 (Remarks to the Author):

The authors show that SAMHD1 is dephosphorylated at G1 of the cell cycle by PP2A. As the dephosphorylated form is the one that restricts HIV, the finding suggests that restriction is different at different stages of the cell cycle and that it is regulated by PP2A. SAMHD1 is found to associate in the cell with specific PP2A trimers containing the B55a subunit. Inhibition of PP2A with okadaic acid result in hyperphosphorylated SAMHD1 that is less restrictive to HIV reverse transcription.

The role of phosphorylation of SAMHD1 in controlling restriction activity is poorly understood and thus of interest. Phosphorylation does not have a major impact on dNTP concentration or catalytic activity of SAMHD1. The authors show that dephosphorylation is controlled by PP2A, a finding that is interesting. It would have been helpful if the findings shed light on what phosphorylation does to SAMHD1 to cause it not to restrict infection.

The finding that PP2A is the phosphatase that dephosphorylates T592 is an important advance and the data may be convincing. The experiments are carefully done with required controls and the paper contains a considerable amount of data. Its hard to know if the results are of interest to a general audience as they are more for the specialists in the field. Some of the data are a bit weak, as detailed below and it would help to have them strengthened.

Specific points

1. Fig. 2A. The mass spec list includes few proteins of interest. These are presumably culled from a larger list. Mass Spec hits always consist many artifactual nonspecific interactions that are picked up as a result of the high sensitivity of the instrument. Hits are only real if they do not appear in a negative control pull-down. Were these hits absent in the negative control? The results need to be shown. The list of top hits for the data should be shown in supplementary data.

2. Fig. 2C and D. The fact that all mutated SAMHD1s still interact with PP2A is problematic and casts doubt on the specificity of these pull-downs. The only mutant that is said not to bind PP2A is R451E, yet that data is not convincing due to the lack of stability of the the mutated protein. Quantification suggests that R451 doesn't bind but looking at the gel suggests otherwise. R451 is poorly expressed yet still pulls down PP2A detectably, suggesting that the interaction is maintained.

3. Fig. 4A. B55a appears to specifically associate with SAMHD1 but this may be a misleading result as the lysate blot shows that this subunit was the most highly expressed. Given that the band is faint, this result may not be correct.

4. Fig. 4A. Why are heavy and light chains of the first antibody not showing up on the gel? The second antibody should have strongly detected the anti-GST antibody. It would be

helpful to explain.

5. Page 8, line 209 and 211. Incorrect reference to Fig. 5B. Should be labeled Fig. 6B.

6. Fig. 6C. There are several background bands making it hard to know that the indicated band is really SAMHD1. Why does this blot have so many bands where the others are so clean? As the background bands disappear more slowly in the siB55a lanes, this would seem to indicate that the SAMHD1 effect is nonspecific.

7. Fig. 7B. It does not appear that the B55a knock down worked. Quantification suggests that it did, but this is not convincing.

8. Fig. 7D. The results are problematic for two reasons. First, OA does not significantly cause an increase in the viral DNA copies. While the authors claim it does, it does not. This would have been an important result as it would show that PP2A affects SAMHD1 restriction activity in the cell. Second, the authors used a luciferase reporter virus yet do not show the data for luciferase activity in the infected cells. Had the OA treatment resulted in higher luciferase activity, this would also confirm the main finding of the paper. The authors need to show the luciferase data.

Reviewer #2 (Remarks to the Author):

Dephosphorylation of residue T592 of the protein SAMHD1 promotes an anti-viral state of cells that restricts infection by HIV-1 and other viruses. Presumably this anti-viral state reflects SAMHD1's dNTPase activity (lowering dNTP pools below levels needed to support reverse transcription, but other purported activities of SAMHD1 (single-stranded nucleic acid binding and an RNA exonuclease function) might also be involved. It is also not clear how the phosphorylation state of T592 influences the anti-viral function of SAMHD1. In this manuscript, the authors show through the combination of several methods that the major phosphatase activity that removes the pT592 phosphorylation and thus activates the anti-viral state is the holoenzyme PP2A-B55.

The evidence for this conclusion is as follows:

- (i) Various kinds of pull-down/co-IP experiments indicate physical interactions between SAMHD1 and PP2A-B55 subunits. It appears using truncation mutants that the C-terminal region of SAMHD1 mediates most of this binding, although the binding sites were not analyzed in detail.
- (ii) pT592 can be dephosphorylated in vitro by a somewhat artefactual PP2A heterodimer, but not by PP1. More convincingly, this reaction is supported by semi-purified PP2A-B55 but not by other PP2A-containing heterotrimers.
- (iii) In cells treated with okadaic acid (an inhibitor of PPP family phosphatases) and in cells depleted of PP2A-B55 by RNAi against its component subunits, levels of pT592 are increased. Okadaic acid treatment of cells also augments HIV-1 infectivity as measured by the appearance of viral products.

Two other aspects of the work are of interest. First, in cycling cells it appears that the activity of PP2A-B55 in dephosphorylating is mostly restricted to the M phase exit/early G1. That is, levels of pT592 are uniformly high until this stage and then decrease. Furthermore, the removal of various M phase phosphorylations as well as the removal of pT592 are delayed after CDK1 inhibition in B55-depleted cells. Second, interferon and Sendai virus treatments strongly upregulate B55 mRNAs expression, although upregulation of the B55 protein is much more modest. This may represent a new and unanticipated level of antiviral activity by interferons, although the manuscript just presents this one result without investigating it further.

My evaluation of this paper with respect to the results reported is mostly positive. Specific experiments could have been performed with greater care in ways that could have been informative. For example, the dephosphorylation of pT592 by PP2A-B55 is time- and concentration-dependent (Fig. 5A), but insofar as I can see, no effort was made to assess the purity or concentration of the enzyme, much less to determine any kinetic constants (K_m and k_{cat}) that would allow some kind of quantitative understanding of this reaction. One could complain about the controls of the RNAi experiment in that (again, insofar as I can see) no effort was made to knock down phosphatases other than PP2A-B55, such as PP1 or other PP2A regulatory subunits. (By the way, in the text this experiment is referenced as Figure 5 when this clearly should be Figure 6. I have not systematically checked all the figure calls in the text, but the authors should certainly do so.)

Several experiments depend on the use of okadaic acid, which is an imperfectly specific inhibitor of PP2A. Other important enzymes like PP4 and PP6 are inhibited by this drug with approximately the same IC_{50} . Furthermore, although use of nanomolar okadaic acid is generally taken to be indicative of PP2A-like enzyme activity, this test has never completely made sense as the concentrations of these enzymes in the cell must be much higher (certainly greater than 10 nM and probably more like 100 nM. Thus, the drug must in some fashion be concentrated in cells relative to the environment in order to inhibit enough of these enzymes to make a difference. Although the okadaic acid IC_{50} s for PP2A and PP1 are $\sim 100X$ different using purified enzymes, this concentration effect means that the discrimination of these enzyme activities in intact cells by okadaic acid treatment is not so clear-cut as the authors would have us believe. Fortunately, with the exception of Figure 7C/D, most of the experiments using okadaic acid in intact cells were backed up by alternative methods using RNAi depletion.

In spite of these quibbles about individual experiments, I believe that sufficient self-consistent data has been accumulated to accept the main thesis of the manuscript, which is that PP2A-B55 is the major phosphatase targeting pT592. This conclusion is of some significance because of the importance of this phosphosite for the success of viral infections. I could have imagined several experiments to have been performed better, but I personally would not make publication of this manuscript contingent on additional experimentation.

The writing is workman-like though hardly elegant. My major objection to the manuscript as it is concerns the Discussion, which was rambling, confusing, and (for the most part) boring.

The final paragraph is particularly useless.

A larger concern is that the authors neglected an extensive literature relating to one of the main points they raise in the manuscript: namely, that PP2A-B55 targets pT592 during mitotic exit. A conserved and well characterized pathway operating through Greatwall kinase and Endosulfine proteins shuts turns off PP2A-B55 function against its normal mitotic substrates during M phase, at least in most cell types. At M phase exit, PP2A-B55 activity against these substrates is reactivated because the phosphatase inactivates the Endosulfine inhibitor. This pathway explains the author's findings in large part. That is, pT592 levels should be high during M phase (because CDK activity is high and PP2A-B55 activity is low), whereas at M phase the situation is reversed and pT592 levels should fall. What is not explained is the elevated phosphorylation during S and G2 phases; presumably CDK2 activity is high during that time although I don't know of any data concerning PP2A-B55 activity at that time. Still, this literature is obviously of importance to the interpretation of this manuscript and in many ways validates the main conclusion. It is unaccountable to me that this was not discussed.

A recent paper in this literature (Cundell et al.; DOI: 10.1083/jcb.201606033) has uncovered the amino acid sequence determinants for PP2A-B55 and related these sequences to the temporal order of protein dephosphorylation during mitotic exit. The authors need to determine whether the region in the vicinity of pT592 fits these sequence requirements. Furthermore, the supporting information for the Cundell et al. paper may already have provided useful confirmatory evidence for the main point made in the manuscript.

The Discussion raises two genuinely interesting ideas/speculations that should be explored at more length. First, the authors suggest that HIV-1 may exploit the cell cycle regulation of PP2A-B55 by expressing the Vpr protein, which arrests infected cells in G2. This could be useful if PP2A-B55 is turned off in these arrested cells (even though they have not reached a classical M phase) so that the virus could evade this anti-viral mechanism. Second is the strong upregulation of B55 by interferons, which could enhance the anti-viral state. Unfortunately, these ideas are not well developed and they get lost in the rambling Discussion. I am not sure if the journal allows subheadings within the Discussion, but if so, they would help the general organization and provide some emphasis for paragraphs dealing with these fascinating points.

In my opinion, this work is potentially publishable in Nature Communications, but only after the Discussion is completely rewritten to address the concerns above

Reviewer #3 (Remarks to the Author):

In this manuscript, Schott et al report that the retroviral restriction factor SAMHD1 is dephosphorylated at T592 by PP2A during mitotic exit in cycling cells. They also find that this regulation is prominent in non-cycling monocyte-derived macrophages (MDMs). Authors show that PP2A preferentially binds multimerized SAMHD1 irrespective of the

phosphorylation status. Targeted knockdown of PP2A-B55a or the treatment of PP2A-inhibitor okadaic acid results in increased amount of pT592 of SAMHD1 thereby potentially restoring the permissiveness for HIV-1 infection. Moreover, the authors show that SAMHD1 phosphorylation at T592 is reduced following the treatment of IFNs probably due to the induction of PP2A-B55a. Since the phosphorylation state of SAMHD1 on this site is crucial for the anti-viral activity, the current study proposes an important regulatory mechanism of SAMHD1-mediated viral restriction.

Overall, this study is well performed and technically sound with solid data in biochemical analysis. However, the cell types that are best studied in this manuscript are predominantly HeLa 'Kyoto' and HEK293T epithelial cells. Also, studies with MDMs are not well designed with proper controls so that their overall hypothesis is not fully clarified. The manuscript would be strengthened by studies to show whether PP2A could facilitate viral restriction via dephosphorylation of SAMHD1 in activated/resting CD4+ T cells as well as cycling/non-cycling monocytic cells. My specific comments were listed below.

1) Figure 1A: Authors synchronized HeLa 'Kyoto' cells at G1/S boundary and released for monitoring SAMHD1-pT592 in line with cyclin/CDK immunoblots. It is better to monitor them until SAMHD1 is re-phosphorylated up to G1/S boundary. Also, phosphorylated Cdc25, another target of PP2A, should be examined in the same blot sets.

2) Figure 2A: The description of procedure for TAP-SAMHD1-interacting proteins is insufficient. Please show gel image for which gel bands are excised for MS analysis. Also, the logic for the selection of listed proteins could be better explained. The authors should include a supplementary table of all SAMHD1-interacting proteins identified by MS and their spectral count numbers. This will be of interest to readers. It would be better to present what proteins were identified in control TAP tag pulldowns.

3) Figure 2C: It is important to analyze whether PP2A directly (or indirectly) binds SAMHD1 and to determine the PP2A-binding region within SAMHD1. I am wondering why both T592E and T592A mutants can still bind PP2A to the same extent as wild-type SAMHD1. Figure 2D: Given that the oligomerization of SAMHD1 is beneficial for binding PP2A (page 5), the R451E mutant may not be dephosphorylated at T592 by PP2A.

4) Figure 3B: SAMHD1 blot is very hard to see and should be amended.

5) Figure 4A: IP products of GST-tagged PP2A-B-type subunits are too variable to evaluate their interaction with SAMHD1. Furthermore, I am very wondering why amounts of co-immunoprecipitated PP2A A- and C-type subunits are variable among different B-type subunits. Also, authors should investigate whether endogenous SAMHD1 binds PP2A in HIV-1 susceptible cells (monocytic cells or CD4+ T cells).

6) Figure 6A: The cyclin B1 blot has only 6 lanes while other blots have 7 lanes. Figure 6B: total SAMHD1 amounts are very variable that should be amended.

7) Figure 6C: Quantification data need statistical analysis with better presentation. It may be also important to examine the increase in dNTP levels along with SAMHD1

phosphorylation.

8) Figure 7: My main concern is referred to the HIV infection-related side of the article. While the increased effect of the PP2A on SAMHD1 dephosphorylation is clear to me, results involving HIV-1 permissiveness are only obtained in MDM cultures, so the relevance for HIV biology is limited or inexistent. The authors should test their hypothesis using HIV-1-infected primary CD4+ T cells and cycling/non-cycling monocytic cells such as THP-1 or SAMHD1-transduced U937. Furthermore, authors only used VSV-G-pseudotyped single-round HIV infection model with non-cycling MDMs. Multi-round HIV-1 infection model could be tested.

9) Figure 7C, D: The magnitude of the influence of OA treatment on viral replication step(s) is very modest. Authors may measure the luciferase activity in the parallel experiment. Previous reports demonstrated that human primary MDMs have only non-phosphorylated SAMHD1. However, authors show that MDMs (especially Donor 2) exhibit very high level of pT592 probably due to improper purification/differentiation of MDMs. Cell surface markers and cell-cycle profile could be analyzed by FACS analysis.

10) Figure 7E: The increased expression of PP2A-B55a in response to IFN treatments has not been mechanistically explored and could occur for a number of reasons, either direct or indirect. Does HIV infection induce PP2A expression? Moreover, it is not clear whether the increased expression of PP2A would affect the viral restriction in MDMs from the presented data. PP2A (wild-type, T592A, T592E)-transduced U937 cells could be useful for analyzing the function of IFN-induced PP2A in terms of HIV-1 infection. A more productive approach could be to more fully explore the exact role of the IFN-PP2A-SAMHD1 axis in HIV-1 replication.

11) Figure 7F: PP2A has multiple substrates so that endogenous one may not efficiently dephosphorylate SAMHD1 in cells. This can be analyzed by prior introduction of siRNA against B55a followed by IFN treatment. Also, it is plausible that an adaptor protein (ISG?) might modulate the interaction between PP2A and SAMHD1 as is often the case (e.g. Okamoto et al., Mol Cell. 2002 9:761-71).

12) T592 is located adjacent to Vpx binding region. Does the interaction of PP2A with SAMHD1 affect the accessibility of Vpx to SAMHD1? Or, conversely, Vpx binding to SAMHD1 affects the PP2A-mediated dephosphorylation of SAMHD1? This could be tested in the presence of proteasomal inhibitors.

13) Supplementary Figure 1C, right panel: Statistical analysis should be needed.

14) Supplementary Figure 2C: SAMHD1 blot looks very messy and should be amended.

15) This reviewer suggests that authors generate summarized cartoon of suggested mechanism.

We would like to thank the reviewers for their valuable feedback. We truly appreciate their critiques, and have made revisions and included additional experimental studies (**new Figures 1c, 2a, 4d-f, 6b, 7a-e, 8a and b; new Supplementary Figures 1a-f, 2a and b, 3a-d, 4e and f, 5e, 6a-e, 7a-c**) to address their concerns. We feel that these revisions have significantly strengthened this manuscript and have further substantiated the conclusions of the study. A point-by-point response is provided below.

Reviewer #1:

The authors show that SAMHD1 is dephosphorylated at G1 of the cell cycle by PP2A. As the dephosphorylated form is the one that restricts HIV, the finding suggests that restriction is different at different stages of the cell cycle and that it is regulated by PP2A. SAMHD1 is found to associate in the cell with specific PP2A trimers containing the B55a subunit. Inhibition of PP2A with okadaic acid result in hyperphosphorylated SAMHD1 that is less restrictive to HIV reverse transcription.

The role of phosphorylation of SAMHD1 in controlling restriction activity is poorly understood and thus of interest. Phosphorylation does not have a major impact on dNTP concentration or catalytic activity of SAMHD1. The authors show that dephosphorylation is controlled by PP2A, a finding that is interesting. It would have been helpful if the findings shed light on what phosphorylation does to SAMHD1 to cause it not to restrict infection.

The finding that PP2A is the phosphatase that dephosphorylates T592 is an important advance and the data may be convincing. The experiments are carefully done with required controls and the paper contains a considerable amount of data. Its hard to know if the results are of interest to a general audience as they are more for the specialists in the field. Some of the data are a bit weak, as detailed below and it would help to have them strengthened.

We thank the reviewer for appreciating our findings. We feel that the revisions based on suggestions of the reviewer (as in the point-by-point response) have significantly strengthened this manuscript and substantiated the conclusions of the study.

Specific points

- 1. Fig. 2A. The mass spec list includes few proteins of interest. These are presumably culled from a larger list. Mass Spec hits always consist many artifactual nonspecific interactions that are picked up as a result of the high sensitivity of the instrument. Hits are only real if they do not appear in a negative control pull-down. Were these hits absent in the negative control? The results need to be shown. The list of top hits for the data should be shown in supplementary data.*

The presented mass spectrometry experiment in the previous version of the manuscript (presented now in **Supplementary Fig. 2a-c**) was based on an in-gel digestion protocol on excised bands. As in-gel digestion protocols are inherent to loss of extracted peptides (e.g. in control excised bands), we have additionally performed and included a second independent mass spectrometry experiment (**new Fig. 2a**) to validate the first mass spectrometry experiment and to prove that the presented hit, PP2A, is a specific interactor. We now used a different tag (GFP instead of CBP-SBP) to confirm the specificity of PP2A-pulldown in comparison to negative control-

pulldowns. Moreover, we used on-bead (in-solution) digestion to prepare the samples (anti-GFP-SAMHD1 trapped interactome and control anti-GFP-trap; **new Fig. 2a**), instead of the previously used in-gel digestion on excised bands (**Supplementary Fig. 2a-c**). The on-bead approach substantiates PP2A as a hit (present in the SAMHD1-pulldown, but absent in control-pulldown). We explain the method of selecting the hits and the selected threshold in more detail in the revised version (**lines 617-625**). In short, only peptides with 95 % identification probability, resulting in a 1 % peptide FDR, were taken into account. As a result, a list of 280 proteins present in the GFP-SAMHD1 trap and not present in the GFP-control trap could be established. CDK1 and the A α subunit of PP2A were found to be genuine interaction partners that are absent from the negative control list. Additionally, they were detected in both mass spectrometry approaches. As these are the relevant proteins for phosphorylation at T592 (the known phosphorylating kinase and PP2A as subject of the current manuscript), we present both in Fig. 2a. The specificity of the interaction between SAMHD1 and PP2A was confirmed in CoIP experiments, even on endogenous proteins (**new Fig. 4d**), and by analyzing the binding motif where we identified residues that abrogate binding further validating the interaction (**Fig. 4e, f**). As the hit list of 280 proteins was not additionally validated and would not add relevant information for the current scope of this manuscript, we would prefer to present only the truly validated interaction partners relevant for phosphorylation at T592.

- 2. Fig. 2C and D. The fact that all mutated SAMHD1s still interact with PP2A is problematic and casts doubt on the specificity of these pull-downs. The only mutant that is said not to bind PP2A is R451E, yet that data is not convincing due to the lack of stability of the the mutated protein. Quantification suggests that R451 doesn't bind but looking at the gel suggests otherwise. R451 is poorly expressed yet still pulls down PP2A detectably, suggesting that the interaction is maintained.*

We agree with the reviewer. In order to assess the specificity of the interaction between PP2A and SAMHD1, we performed additional experiments. We took advantage of the recently described motif specific to substrates of the PP2A-B55 holoenzyme (Cundell *et al.*, 2016; PMID: 27551054). Cundell *et al.* describe a defined bipartite polybasic motif flanking the CDK recognition site in question. Interestingly, when comparing basic residues (from aa551 to 626) in human SAMHD1 to the corresponding residues in murine SAMHD1 (**new Fig. 4e**), we identified 6 basic sites in human SAMHD1 as non-basic in mouse SAMHD1. In fact, we were able to detect an almost complete loss of PP2A binding to mouse SAMHD1, which could be reverted by exchanging the corresponding amino acids to basic residues and vice versa (**new Fig. 4f**). We believe that these residues strongly influence binding to PP2A and demonstrate specificity of the interaction.

Consequently, we moved previous Fig. 2C, D to Supplementary Fig. 2d, e. Particularly with regard to the new findings in conjunction with the structural model of

SAMHD1 (**new Supplementary Fig. 7**), we discussed the position of the key residues for binding to PP2A (**lines 418-426 of the revised manuscript**). Therefore, we cautioned our statement on the R451E mutant (**lines 124-128**) as we agree with the reviewer that this experiment was not sufficient to convincingly draw this conclusion.

3. *Fig. 4A. B55a appears to specifically associate with SAMHD1 but this may be a misleading result as the lysate blot shows that this subunit was the most highly expressed. Given that the band is faint, this result may not be correct*

We apologize if the figure might have been misleading. In fact, GST-tagged B55 α is the least expressed subunit in the lysate in comparison to other B-type subunits (**Fig. 4a, right panel**). In order to avoid misinterpretation, we explained the figure in greater detail in the revised version (**lines 169-174**). We wish to emphasize that we performed the reciprocal pulldown of SAMHD1 to demonstrate that B55 α is interacting with SAMHD1 with high affinity (**Fig. 4b, c**). Indeed, we cannot completely exclude binding of other subtypes (as there are 23 isoforms). It is likely that other subtypes may interact with SAMHD1 with lower affinity and target other phosphorylation sites. But we would like to emphasize that we validated B55 α by *in vitro*-dephosphorylation experiments to prove that B55 α is the responsible B-type subunit leading to dephosphorylation at the specific site T592 (**Fig. 5c and Supplementary Fig. 4e**). PP2A-B56 α , PP2A-B56 β and PP2A-PR72 holoenzymes which did not bind to SAMHD1 in cells with high affinity (**Fig. 4a**) were also not able to remove T592 phosphorylation *in vitro* (**Fig. 5c and Supplementary Fig. 4e**). Therefore, we believe, this underscores specific SAMHD1-PP2A-B55 α interactions relevant for the T592 site.

4. *Fig. 4A. Why are heavy and light chains of the first antibody not showing up on the gel? The second antibody should have strongly detected the anti-GST antibody. It would be helpful to explain*

We now explain our method in more detail in the revised manuscript (**lines 642-646**): GST-Trap_M beads are based on a second type of antibody of the *Camelidae* species (a single variable domain (VHH) devoid of light chains). As there are no heavy and light chains that would interfere with downstream applications, such as ColIPs or mass spectrometry, we have used these *Camelidae* antibodies in this experiment.

5. *Page 8, line 209 and 211. Incorrect reference to Fig. 5B. Should be labeled Fig. 6B.*

The incorrect label got changed. Previous Fig. 6B is now moved to **new Fig. 5e**.

6. *Fig. 6C. There are several background bands making it hard to know that the indicated band is really SAMHD1. Why does this blot have so many bands where the others are so clean? As the background bands disappear more slowly in the siB55a lanes, this would seem to indicate that the SAMHD1 effect is nonspecific.*

Since the phosphorylation site T592 in SAMHD1 represents the perfect CDK consensus sequence, it is very likely that the SAMHD1 phospho-antibody may recognize other substrates, especially in mitotic samples where phosphorylated CDK substrates are accumulating. Since the mitotic exit assay is designed to particularly enrich for CDK substrates and display dephosphorylation of all kind of CDK substrates by chemical induction of mitotic exit (see also **new Supplementary Fig. 5d**), we believe the antibody does recognize other proteins besides SAMHD1. The assay was first developed by Schmitz *et al.*, 2010 (PMID: 20711181, Fig. 4h; **Supplementary Fig. 5** in this publication). By using this assay, we demonstrate that silencing of B55 α leads to a delay in dephosphorylation of SAMHD1 (**new Fig. 6a**) besides other CDK substrates (**new Supplementary Fig. 5d**), verifying PP2A-B55 α trimer as the mitotic exit phosphatase for SAMHD1. In order to designate the correct SAMHD1 band, we compared samples of mitotic cells to non-synchronized HeLa 'Kyoto' cells side-by-side (see figure supporting rebuttal 1 below). We indicated this approach in the figure legend of **new Fig. 6a**.

7. *Fig. 7B. It does not appear that the B55a knock down worked. Quantification suggests that it did, but this is not convincing.*

The presentation of the figure might have been unclear. In fact, right panel of the figure (now **new Fig. 5g, right panel**) does not represent the quantification of the left panel. Instead, the left side shows the western blot detection of PP2A B55 α protein level (**new Fig. 5g, left panel**). The right graph represents the fold change of PP2A B55 α mRNA levels (**new Fig. 5g, right panel**). Now, we labeled the Y-axis of the graph with "fold change *PPP2R2A* (B55 α) mRNA" and added the quantification of the western blot in **new Fig. 5g, left panel**. Still, highly efficient silencing of B55 α mRNA (**Fig. 5g, right panel**) translates only in a modest downregulation of B55 α subunit protein expression (**Fig. 5g, left panel**) (**lines 246-250**). This is very likely attributable to a long target protein half-life. In general, subunits of PP2A have been shown to be highly stable, e.g. C α subunits of PP2A are specifically protected from degradation by other cellular proteins (Yabe *et al.*, 2015, PMID: 26678046; Kong *et al.*, 2009, PMID: 19818709)

8. *Fig. 7D. The results are problematic for two reasons. First, OA does not significantly cause an increase in the viral DNA copies. While the authors claim it does, it does not. This would have been an important result as it would show that PP2A affects SAMHD1 restriction activity in the cell. Second, the authors used a luciferase reporter*

virus yet do not show the data for luciferase activity in the infected cells. Had the OA treatment resulted in higher luciferase activity, this would also confirm the main finding of the paper. The authors need to show the luciferase data.

We agree with the reviewer that the infectivity data on MDMs were not satisfying. We realized that upon inhibition of PP2A, the effect of dephosphorylation of SAMHD1 on HIV infection is opposed by other cellular functions of PP2A affecting HIV replication, directly or indirectly. We outlined the main contributing causes as why the detection window might only allow a minor effect upon infection in MDMs: (i) inhibition of PP2A induces an antiviral state in immune-competent cells, such as THP-1 or MDMs, counteracting the effect of PP2A on SAMHD1; (ii) inhibition of PP2A has been shown to affect many cellular processes; in particular, it inhibits CDK2 activity, which is necessary for proper RT activity. As pointed out below, we determined sensing-competent cells, such as MDMs, as an unsuitable model to monitor infection upon inhibition of PP2A. Therefore, we excluded the results (previous Fig. 7C, D) from the revised version. Instead, we decided to set the focus of the manuscript on the effect of PP2A in proliferating cells and present now an alternative infection model on proliferating cells in **new Fig. 7**: Even in cycling cells, in the G₁ stage of the cell cycle, SAMHD1 dephosphorylation (when PP2A activates SAMHD1 (**new Fig. 6a, b**) and coincident with a drop in dATP levels (**new Fig. 1c, 6b**) leads to a significant decrease in HIV-1 late RT products, whereas RT products are not affected when SAMHD1 is not present - suggesting that during the G₁ phase SAMHD1 is an active enzyme inhibiting HIV-1 (**new Fig. 7, cartoon Fig. 8**).

- i) We established RNAi inhibition of PP2A in MDMs (see figure supporting rebuttal 2a and b, left panel). Upon knockdown of single subunits (B55 α or C α) or triple knockdown of A α +B α +C α , as expected, phosphorylation at T592 increased. We observed, though, upregulation of the interferon pathway upon silencing of PP2A (as indicated by upregulation of STAT1). Recently, PP2A has been demonstrated to negatively regulate the IRF3 pathway upon stimulation or Sendai virus infection. IRF3 is then dephosphorylated with the consequence of suppressing VSV replication, as shown in Long *et al.* (Long *et al.*, 2014; PMID: 24726876). In fact, flaviviruses are known to utilize the negative regulatory role of PP2A for their own advantage and induce PP2A activity in order to block IFN-mediated restrictions (Duong *et al.*, 2005, PMID: 16306605; Duong *et al.*, 2004, PMID: 14699505). Additionally, PP2A depletion using siRNAs was shown to increase levels of pro-inflammatory cytokines *in vitro* (Cornell *et al.*, 2009, PMID: 19286927; Sun *et al.*, 2007, PMID: 17170118). We observed upregulation of STAT1 upon silencing of PP2A indicative of a host antiviral response. Consequently, subsequent infection experiments mirrored the antiviral state of the host cell counteracting the effect of upregulated SAMHD1 phosphorylation in MDMs. Moreover, an antibody cocktail to block IFN (α -IFN α , α -IFN β and α -IFNAR) (see figure supporting rebuttal 2b, right panel) blocked STAT1 activation, but unfortunately

affected the phosphorylation levels by an unknown mechanism, therefore prevented us from using this cocktail. Therefore, we determined sensing-competent cells, such as MDMs, as an unsuitable model to monitor infection.

- ii) Additionally, inhibition of PP2A (e.g. through chemical inhibition by OA) has been shown to affect many cellular processes; in particular, it inhibits CDK2 activity (Yan *et al.*, 1999; PMID: 10542219). As effective HIV-1 reverse transcription is known to be dependent on phosphorylation of the reverse transcriptase (RT) protein by CDK2 (Leng *et al.*, 2014; PMID: 24922574), inhibition of PP2A might inhibit effective RT and affect the stability of RT. Therefore, altered HIV-1 infectivity in MDMs mirroring the increase in SAMHD1 phosphorylation at T592 would be opposed by less efficient RT due to dephosphorylation of the RT protein. We believe that this point together with the antiviral state induced by IFN might be the major contributors explaining the minor effect we detect upon infection in MDMs.
- iii) As suggested by the reviewer, we used a luciferase reporter virus: In contrast to RT products upon OA inhibition (as in previous Fig. 7C, D), luciferase expression is massively decreased upon OA treatment (see figure supporting rebuttal 3 below). Again, we believe, most likely an IFN-mediated block which acts past RT might affect luciferase expression. Additionally, indirect effects (OA is known to inhibit protein synthesis; e.g. Matias *et al.*, 1996, PMID: 8656452) or direct effects (CDK9/ CDK2 inhibition due to PP2A inhibition; Breuer *et al.*, 2012, PMID: 23140174) might be responsible for the massive drop in expression levels. Therefore, these additional effects, inherent to the use of OA, prevented us to analyze luciferase gene expression.

Reviewer #2:

Dephosphorylation of residue T592 of the protein SAMHD1 promotes an anti-viral state of cells that restricts infection by HIV-1 and other viruses. Presumably this anti-viral state reflects SAMHD1's dNTPase activity (lowering dNTP pools below levels needed to support reverse transcription, but other purported activities of SAMHD1 (single-stranded nucleic acid binding and an RNA exonuclease function) might also be involved. It is also not clear how the phosphorylation state of T592 influences the anti-viral function of SAMHD1. In this manuscript, the authors show through the combination of several methods that the major phosphatase activity that removes the pT592 phosphorylation and thus activates the anti-viral state is the holoenzyme PP2A-B55.

The evidence for this conclusion is as follows:

(i) Various kinds of pull-down/co-IP experiments indicate physical interactions between SAMHD1 and PP2A-B55 subunits. It appears using truncation mutants that the C-terminal region of SAMHD1 mediates most of this binding, although the binding sites were not analyzed in detail.

(ii) pT592 can be dephosphorylated *in vitro* by a somewhat artefactual PP2A heterodimer, but not by PP1. More convincingly, this reaction is supported by semi-purified PP2A-B55 but not by other PP2A-containing heterotrimers.

(iii) In cells treated with okadaic acid (an inhibitor of PPP family phosphatases) and in cells depleted of PP2A-B55 by RNAi against its component subunits, levels of pT592 are increased. Okadaic acid treatment of cells also augments HIV-1 infectivity as measured by the appearance of viral products

Two other aspects of the work are of interest. First, in cycling cells it appears that the activity of PP2A-B55 in dephosphorylating is mostly restricted to the M phase exit/early G1. That is, levels of pT592 are uniformly high until this stage and then decrease. Furthermore, the removal of various M phase phosphorylations as well as the removal of pT592 are delayed after CDK1 inhibition in B55-depleted cells. Second, interferon and Sendai virus treatments strongly upregulate B55 mRNA expression, although upregulation of the B55 protein is much more modest. This may represent a new and unanticipated level of antiviral activity by interferons, although the manuscript just presents this one result without investigating it further.

My evaluation of this paper with respect to the results reported is mostly positive. Specific experiments could have been performed with greater care in ways that could have been informative. For example, the dephosphorylation of pT592 by PP2A-B55 is time- and concentration-dependent (Fig. 5A), but insofar as I can see, no effort was made to assess the purity or concentration of the enzyme, much less to determine any kinetic constants (K_m and k_{cat}) that would allow some kind of quantitative understanding of this reaction. One could complain about the controls of the RNAi experiment in that (again, insofar as I can see) no effort was made to knock down phosphatases other than PP2A-B55, such as PP1 or other PP2A regulatory subunits. (By the way, in the text this experiment is referenced as Figure 5 when this clearly should be Figure 6. I have not systematically checked all the figure calls in the text, but the authors should certainly do so.)

Several experiments depend on the use of okadaic acid, which is an imperfectly specific inhibitor of PP2A. Other important enzymes like PP4 and PP6 are inhibited by this drug with approximately the same IC_{50} . Furthermore, although use of nanomolar okadaic acid is generally taken to be indicative of PP2A-like enzyme activity, this test has never completely made sense as the concentrations of these enzymes in the cell must be much higher (certainly greater than 10 nM and probably more like 100 nM. Thus, the drug must in some fashion be concentrated in cells relative to the environment in order to inhibit enough of these enzymes to make a difference. Although the okadaic acid IC_{50} s for PP2A and PP1 are ~100X different using purified enzymes, this concentration effect means that the discrimination of these enzyme activities in intact cells by okadaic acid treatment is not so clear-cut as the authors would have us believe. Fortunately, with the exception of Figure 7C/D, most of the experiments using okadaic acid in intact cells were backed up by alternative methods using RNAi depletion.

In spite of these quibbles about individual experiments, I believe that sufficient self-consistent data has been accumulated to accept the main thesis of the manuscript, which is that PP2A-B55 is the major phosphatase targeting pT592. This conclusion is of some significance because of the importance of this phosphosite for the success of viral infections. I could have

imagined several experiments to have been performed better, but I personally would not make publication of this manuscript contingent on additional experimentation.

We appreciate the reviewer's positive evaluation. Even so the reviewer pointed out that additional experimentation may not be necessary, we added several additional datasets to support our conclusion. New datasets were added to **new Figures 1c, 2a, 4d-f, 6b, 7a-e, 8a and b; new Supplementary Figures 1a-f, 2a and b, 3a-d, 4e and f, 5e, 6a-e, 7a-c.**

Specifically, we appreciated the reviewer's detailed insight into the amino acid sequence determinants for PP2A-B55 target proteins. In order to assess the specificity of the interaction between PP2A and SAMHD1, we took advantage of the recently described motif, a defined bipartite polybasic motif flanking the CDK recognition site, specific to substrates of the PP2A-B55 holoenzyme (Cundell *et al.*, 2016; PMID: 27551054). Interestingly, when comparing basic residues (from aa551 to 626) in human SAMHD1 to the corresponding residues in murine SAMHD1 (**new Fig. 4e**), we identified 6 basic sites in human SAMHD1 as non-basic in mouse SAMHD1. In fact, we were able to detect an almost complete loss of PP2A binding to mouse SAMHD1, which could be reverted by exchanging the corresponding amino acids to basic residues and vice versa (**new Fig. 4f**). Hence, we believe, that these residues strongly influence binding to PP2A and, at the same time, demonstrate specificity of the interaction.

In order to further characterize the recognition motif, we set out to mutate all or some of the identified positively charged residues in human SAMHD1 surrounding the CDK motif (see figure supporting rebuttal 4a, b below), thereby creating a set of six different mutants of SAMHD1. Unfortunately, only 2 of the mutants were well expressed. The exchanges in mutant 2 (⁵⁹⁵AA) and mutant 4 (⁵⁹⁵AA including mutations of positively charged residues downstream the CDK site) were not sufficient to clarify the exact binding motif, and unfortunately, protein destabilization by other mutations prevented us from analyzing this further. Still, we identified the six sites as key residues that abrogate binding to PP2A (**new Fig. 4e, f**), therefore, we decided to include this panel in the manuscript.

Particularly, based on the new findings in conjunction with the structural model of SAMHD1 (**new Supplementary Fig. 7**), we discussed the position of the key residues for binding to PP2A (**lines 418-426 of the revised manuscript**).

The writing is workman-like though hardly elegant. My major objection to the manuscript as it is concerns the Discussion, which was rambling, confusing, and (for the most part) boring. The final paragraph is particularly useless.

A larger concern is that the authors neglected an extensive literature relating to one of the main points they raise in the manuscript: namely, that PP2A-B55 targets pT592 during mitotic exit. A conserved and well characterized pathway operating through Greatwall kinase and Endosulfine proteins shuts turns off PP2A-B55 function against its normal mitotic

substrates during M phase, at least in most cell types. At M phase exit, PP2A-B55 activity against these substrates is reactivated because the phosphatase inactivates the Endosulfine inhibitor. This pathway explains the author's findings in large part. That is, pT592 levels should be high during M phase (because CDK activity is high and PP2A-B55 activity is low), whereas at M phase the situation is reversed and pT592 levels should fall. What is not explained is the elevated phosphorylation during S and G2 phases; presumably CDK2 activity is high during that time although I don't know of any data concerning PP2A-B55 activity at that time. Still, this literature is obviously of importance to the interpretation of this manuscript and in many ways validates the main conclusion. It is unaccountable to me that this was not discussed.

A recent paper in this literature (Cundell et al.; DOI: 10.1083/jcb.201606033) has uncovered the amino acid sequence determinants for PP2A-B55 and related these sequences to the temporal order of protein dephosphorylation during mitotic exit. The authors need to determine whether the region in the vicinity of pT592 fits these sequence requirements. Furthermore, the supporting information for the Cundell et al. paper may already have provided useful confirmatory evidence for the main point made in the manuscript.

We thank the reviewer for pointing this out. As discussed in the above section, we identified the key residues relevant for PP2A-B55 binding based on the findings presented in Cundell et al. The supplemental tables of Cundell et al. demonstrate the SAMHD1 T592 phospho-site, but the measurements did not seem to meet the criteria for the constrained model to fit neither B55-dependent nor B55-independent dephosphorylation rates.

The Discussion raises two genuinely interesting ideas/speculations that should be explored at more length. First, the authors suggest that HIV-1 may exploit the cell cycle regulation of PP2A-B55 by expressing the Vpr protein, which arrests infected cells in G2. This could be useful if PP2A-B55 is turned off in these arrested cells (even though they have not reached a classical M phase) so that the virus could evade this anti-viral mechanism. Second is the strong upregulation of B55 by interferons, which could enhance the anti-viral state. Unfortunately, these ideas are not well developed and they get lost in the rambling Discussion. I am not sure if the journal allows subheadings within the Discussion, but if so, they would help the general organization and provide some emphasis for paragraphs dealing with these fascinating points.

In my opinion, this work is potentially publishable in Nature Communications, but only after the Discussion is completely rewritten to address the concerns above.

We appreciate the reviewer's constructive suggestions. We thoroughly restructured and revised the discussion and included all the valuable points suggested by the reviewer.

Reviewer #3 (Remarks to the Author):

In this manuscript, Schott et al report that the retroviral restriction factor SAMHD1 is dephosphorylated at T592 by PP2A during mitotic exit in cycling cells. They also find that this regulation is prominent in non-cycling monocyte-derived macrophages (MDMs). Authors show that PP2A preferentially binds multimerized SAMHD1 irrespective of the phosphorylation status. Targeted knockdown of PP2A-B55a or the treatment of PP2A-inhibitor okadaic acid results in increased amount of pT592 of SAMHD1 thereby potentially restoring the permissiveness for HIV-1 infection. Moreover, the authors show that SAMHD1 phosphorylation at T592 is reduced following the treatment of IFNs probably due to the induction of PP2A-B55a. Since the phosphorylation state of SAMHD1 on this site is crucial for the anti-viral activity, the current study proposes an important regulatory mechanism of SAMHD1-mediated viral restriction. Overall, this study is well performed and technically sound with solid data in biochemical analysis. However, the cell types that are best studied in this manuscript are predominantly HeLa 'Kyoto' and HEK293T epithelial cells. Also, studies with MDMs are not well designed with proper controls so that their overall hypothesis is not fully clarified. The manuscript would be strengthened by studies to show whether PP2A could facilitate viral restriction via dephosphorylation of SAMHD1 in activated/resting CD4+ T cells as well as cycling/non-cycling monocytic cells. My specific comments were listed below.

We thank the reviewer for appreciating the importance of addressing the regulatory mechanism and the technical performance of our study. We acknowledged the reviewer's concerns and believe that the additional experiments strengthened the manuscript and substantiated the conclusions of the study as detailed in the point-to-point response.

- 1) *Figure 1A: Authors synchronized HeLa 'Kyoto' cells at G1/S boundary and released for monitoring SAMHD1-pT592 in line with cyclin/CDK immunoblots. It is better to monitor them until SAMHD1 is re-phosphorylated up to G1/S boundary. Also, phosphorylated Cdc25, another target of PP2A, should be examined in the same blot sets.*

We now have fully addressed this point and show two complete rounds of the cell cycle including re-phosphorylation of SAMHD1 upon entry into S phase (**Supplementary Fig. 1d, e**). As suggested, we monitored an additional target of PP2A: Here, we chose a well-known target of B55 α -holoenzymes PRC1 (Cundell *et al.*, 2013; PMID: 24120663) over Cdc25, as this is more likely a target of B56 holoenzymes. As expected, PRC1 resembled the dephosphorylation pattern of SAMHD1 upon entry in G₁ phase (**Supplementary Fig. 1c, f**). Respective samples were split for dNTP measurements (**Fig. 1c**), immunoblotting on SAMHD1 and PRC1 (**Supplementary Fig. 1a, c and Supplementary Fig. 1d, f, respectively**) and propidium iodide staining (**Supplementary Fig. 1b and Supplementary Fig. 1e, respectively**) to determine cell cycle phases by flow cytometry.

- 2) *Figure 2A: The description of procedure for TAP-SAMHD1-interacting proteins is insufficient. Please show gel image for which gel bands are excised for MS analysis. Also, the logic for the selection of listed proteins could be better explained. The authors should include a supplementary table of all SAMHD1-interacting proteins*

identified by MS and their spectral count numbers. This will be of interest to readers. It would be better to present what proteins were identified in control TAP tag pulldowns.

We now included the gel images and the intermediate purification steps separated by SDS-PAGE of the initial mass spectrometry experiment (on CBP-SBP-affinity purified SAMHD1) in **Supplementary Fig. 2a and b**. As the gel-excised bands probably lost extracted peptides and we therefore cannot conclude on absence of PP2A in control excised bands, we decided to undertake a second independent mass spectrometry experiment (**new Fig. 2a**) to validate the first mass spectrometry experiment (presented now in Supplementary Fig. 2). We used a different tag (GFP instead of CBP-SBP) to confirm the specificity of PP2A-pulldown in comparison to negative control-pulldowns. Secondly, we now used on-bead (in-solution) digestion to prepare the samples (anti-GFP-SAMHD1 trapped interactome and control anti-GFP-trap; see materials) instead of the previously used in-gel digestion on excised bands (**Supplementary Fig. 2**). The on-bead approach substantiates PP2A as a hit (present in the SAMHD1-pulldown, but absent in control-pulldown). We explain the method of selecting the hits and the selected threshold in more detail in the revised version (**lines 617-625**). In short, only peptides with 95 % identification probability, resulting in a 1 % peptide FDR, were taken into account. As a result, a list of 280 proteins present in the GFP-SAMHD1 trap and not present in the GFP-control trap could be established. CDK1 and the A α subunit of PP2A were found to be genuine interaction partners that are absent from the negative control list. Additionally, they were detected in both mass spectrometry approaches. As these are the relevant proteins for phosphorylation at T592 (the known phosphorylating kinase and PP2A as subject of the current manuscript), we presented both in **Fig. 2a**. As the hit list of 280 proteins was not additionally validated and would not add relevant information for the current scope of this manuscript, we would prefer to present only the truly validated interaction partners relevant for phosphorylation at T592.

- 3) *Figure 2C: It is important to analyze whether PP2A directly (or indirectly) binds SAMHD1 and to determine the PP2A-binding region within SAMHD1. I am wondering why both T592E and T592A mutants can still bind PP2A to the same extent as wild-type SAMHD1. Figure 2D: Given that the oligomerization of SAMHD1 is beneficial for binding PP2A (page 5), the R451E mutant may not be dephosphorylated at T592 by PP2A.*

We now determined the PP2A-binding region within SAMHD1. Therefore, we believe both proteins interact directly with each other. We took advantage of the recently described motif specific to substrates of the PP2A-B55 holoenzyme (Cundell *et al.*, 2016; PMID: 27551054). Cundell *et al.* describe a defined bipartite polybasic motif flanking the CDK recognition site. This motif rather than the CDK site itself influences binding and dephosphorylation. Interestingly, when comparing basic residues (from aa551 to 626) in human SAMHD1 to the corresponding residues in murine SAMHD1

(**new Fig. 4e**), we identified 6 basic sites in human SAMHD1 as non-basic in mouse SAMHD1. In fact, we were able to detect an almost complete loss of PP2A binding to mouse SAMHD1, which could be reverted by exchanging the corresponding amino acids to basic residues and vice versa (**new Fig. 4f**). We believe, that these residues strongly influence binding to PP2A and demonstrate specificity of the interaction.

Consequently, we moved previous Fig. 2C, D to **Supplementary Fig. 2d, e**. Particularly with regard to the new findings in conjunction with the structural model of SAMHD1 (**new Supplementary Fig. 7**), we discussed the position of the key residues for binding to PP2A (**lines 418-426 of the revised manuscript**). Therefore, we cautioned our statement on the R451E mutant (**lines 124-128**) as we cannot convincingly draw a conclusion.

4) 4) *Figure 3B: SAMHD1 blot is very hard to see and should be amended.*

New Fig. 3b got amended.

5) *Figure 4A: IP products of GST-tagged PP2A-B-type subunits are too variable to evaluate their interaction with SAMHD1. Furthermore, I am very wondering why amounts of co-immunoprecipitated PP2A A- and C-type subunits are variable among different B-type subunits. Also, authors should investigate whether endogenous SAMHD1 binds PP2A in HIV-1 susceptible cells (monocytic cells or CD4+ T cells).*

We now demonstrate binding of endogenously expressed SAMHD1 in THP-1 cells to endogenous PP2A B55 α subunit (**new Fig. 4d**) to further prove relevant interaction in target cells of HIV-1. It is known that PP2A regulation through post-translational modifications change the binding affinity of the core dimer (A&C subunit) toward distinct regulatory B-type subunits and provides specific activity to the holoenzyme (Seshacharyulu *et al.*, 2013, PMID: 23454242; Leulliot *et al.*, 2004, PMID: 14660564); therefore, the binding affinities of different B-type subunits to the core enzyme can be variable (as seen in Fig. 4a). Moreover, there is plenty of evidence in the literature that different B-type subunits, intrinsically, show different binding affinities for the A-C dimer (e.g. Janssens *et al.*, 2003, PMID: 12524438). We wish to emphasize that we performed pulldown of various B-type subunits as well the reciprocal pulldown of SAMHD1 to demonstrate that B55 α is interacting with SAMHD1 with high affinity (**Fig. 4a-c**). Indeed, we cannot completely exclude binding of other subtypes (as there are 23 isoforms). It is likely that other subtypes may interact with SAMHD1 with lower affinity and target other phosphorylation sites. But we would like to emphasize that we validated B55 α by *in vitro*-dephosphorylation experiments to prove that B55 α is the responsible B-type subunit leading to dephosphorylation at the specific site T592 (**Fig. 5c and Supplementary Fig. 4e**).

PP2A-B56 α , PP2A-B56 β and PP2A-PR72 holoenzymes which did not bind to SAMHD1 in cells with high affinity (**Fig. 4a**) were also not able to remove T592 phosphorylation *in vitro* (**Fig. 5c and Supplementary Fig. 4e**). Therefore, we believe, this underscores specific SAMHD1 pT592 dephosphorylation by PP2A-B55 α trimers.

6) *Figure 6A: The cyclin B1 blot has only 6 lanes while other blots have 7 lanes. Figure 6B: total SAMHD1 amounts are very variable that should be amended*

We apologize for the mistake and amended **new Fig. 5d** (former Fig. 6A). Uncropped blots are shown in Figure supporting rebuttal 5 below.

Regarding former Fig. 6B (now **new Fig. 5e**): In fact, we were surprised to observe overall endogenous SAMHD1 protein levels decreasing after PP2A knock-down or inhibition (**Fig. 5e** or upon OA inhibition for 24 h). PP2A silencing leads to hyperphosphorylated SAMHD1, which might be degraded through an unknown mechanism. We noted this in the manuscript in **lines 229-230**. Consequently, hyperphosphorylation of T592 site in SAMHD1 seems not to be tolerated in G₁ phase. Interestingly to note, this is a known mechanism to prevent hyperphosphorylation of proteins in G₁/ S phase, necessary to properly transit through the cell cycle. For instance, in yeast, cells lacking the homologue of PP2A-B55 (Cdc55) contain drastically reduced Cln2 levels caused by E3-ligase-dependent ubiquitination and degradation of the G₁ cyclin homologue Cln2, triggered by an increase in the phosphorylation status of the cyclin (McCourt *et al.*, 2013; PMID: 23518505). In mammalian cells, one example is a member of the retinoblastoma family of proteins: p130 levels are dramatically reduced as cells pass through G₁ phase following its hyperphosphorylation (Bhattacharya *et al.*, 2003; PMID: 12717421). Whether residual phosphorylation at residue T592 and/ or at other phospho-residues in SAMHD1, that are potentially targeted by PP2A, act as degradation signals, needs to be addressed in future studies.

7) *Figure 6C: Quantification data need statistical analysis with better presentation. It may be also important to examine the increase in dNTP levels along with SAMHD1 phosphorylation.*

We now quantified two independent mitotic exit experiments (**Supplementary Fig. 5c**: quantification of two experiments including standard deviations; **new Fig. 6a**: representative of one of those experiments) and demonstrated delayed dephosphorylation of SAMHD1 along delayed mitotic exit upon silencing of B55 α . This provides validation that on-target depletion of B55 α causes prolonged phosphorylation of SAMHD1. We appreciated the reviewer's suggestion on measuring dNTP levels along mitotic exit. Indeed, within 18 min upon flavopiridol addition, dATP levels decreased significantly (**new Fig. 6b**) mirroring the dephosphorylation of SAMHD1 (**new Supplementary Fig. 5e**). MG-132 is constantly

present during these 18 min, preventing degradation of other proteins potentially involved in dNTP metabolism. Therefore, we conclude that SAMHD1 is specifically targeted by PP2A-B55 α holoenzymes during mitotic exit and is converted into an active enzyme lowering dNTP pools, possibly providing an antiviral-active state. Adding additional siRNA-treated samples to test delayed dNTP levels have proven to be extremely challenging: The narrow time window of this assay (taking samples every three minutes) to stop the reactions was a major factor that precluded proper assay outcome. Nevertheless, we present clear evidence of dATP depletion upon dephosphorylation of SAMHD1 within mitotic exit.

- 8) *Figure 7: My main concern is referred to the HIV infection-related side of the article. While the increased effect of the PP2A on SAMHD1 dephosphorylation is clear to me, results involving HIV-1 permissiveness are only obtained in MDM cultures, so the relevance for HIV biology is limited or inexistent. The authors should test their hypothesis using HIV-1-infected primary CD4+ T cells and cycling/non-cycling monocytic cells such as THP-1 or SAMHD1-transduced U937. Furthermore, authors only used VSV-G-pseudotyped single-round HIV infection model with non-cycling MDMs. Multi-round HIV-1 infection model could be tested.*

First, we wish to emphasize the relevance for HIV biology. This work provides critical novel insight into newly identified phosphatase that renders SAMHD1 anti-virally active. Moreover, we defined the exact time window during cell cycle-progression where dephosphorylation happens in parallel to dATP depletion, mirroring the activation of the enzyme (**new Fig. 1c and new Fig. 6b**). We realized that upon inhibition of PP2A, the effect of dephosphorylation of SAMHD1 on HIV infection is opposed by other cellular functions of PP2A affecting HIV replication, directly or indirectly. We outlined the main contributing causes below. Due to the observed challenges using cell models analyzing the inhibition of PP2A on HIV infectivity (see below), we, instead, set the focus of the manuscript on the effect of PP2A in proliferating cells and present now an alternative infection model in **new Fig. 7**: Even in cycling cells, in the G₁ stage of the cell cycle, SAMHD1 dephosphorylation (when PP2A activates SAMHD1 (**new Fig. 6a, b**) and coincident with a drop in dATP levels (**new Fig. 1c, 6b**) leads to a significant decrease in HIV-1 late RT products, whereas RT products are not affected when SAMHD1 is not present - suggesting that during the G₁ phase SAMHD1 is an active enzyme inhibiting HIV-1 (**new Fig. 7, cartoon Fig. 8**).

As suggested by the reviewer, we intended to analyze monocytic cells in order to use them as model cells for infection experiments. We chose two options to perform these experiments: TPA to induce a non-cycling stage or a double-thymidine block/release protocol to get the resting G₁ population. We chose two methods to inhibit PP2A: Okadaic acid or RNAi to silence PP2A. However, we believe there are three main obstacles that precluded us from using these cell models (or even any cell model): (i) phospho-regulation in TPA-differentiated cells does not reflect the

regulation in MDMs; (ii) inhibition of PP2A induces an antiviral state in immune-competent cells, such as THP-1 or MDMs, counteracting the effect of PP2A on SAMHD1; (iii) inhibition of PP2A has been shown to affect many cellular processes; in particular, it inhibits CDK2 activity, which is necessary for proper RT activity. We believe that this point together with the antiviral state induced by IFN might be the major contributors explaining minor effects we detect upon infection.

To (i): We think, the phospho-regulation in TPA-differentiated cells does not reflect the regulation in primary MDMs: first, TPA-differentiated THP-1 cells contain readily detectable levels of phosphorylation (levels are much higher when compared to other primary cells (see **Supplementary Fig. 3b**). We think that more phosphorylated SAMHD1 is present in the mixture of cells in different cell cycle stages after TPA differentiation of THP-1 cells that do not reflect the G₀/G₁-state of MDMs (as indicated by the cyclin A2 and MCM2 expression, see **Supplementary Fig. 3b** and Traore *et al.*, 2005; PMID: 15978937). Traore *et al.* demonstrate that approx. 40 % of TPA-differentiated cells are not in G₀/ G₁. Additionally, when we blocked PP2A activity by OA in THP-1 cells, after 6 h, the cells started to display an unclear phenotype (signal pattern of phosphorylation as seen during mitotic exit, but degradation of cyclin B1) (see figure supporting rebuttal 6b below) and lost at some point SAMHD1 expression (probably similar to the above described phenomenon, that PP2A silencing leads to degradation of hyperphosphorylated SAMHD1).

Secondly, by employing RNAi experiments, either by generating stable knockdown cells (for each subunit A α , C α , C β , B55 α) or by transient transfection with siRNAs, we observed that silencing efficiency was in none of the cases really appreciable resulting in none to marginal differences in phosphorylation, similar what we observed in MDMs (**Fig. 5g**): Highly efficient silencing of B55 α mRNA (**Fig. 5g, right panel**) translates only in a modest downregulation of B55 α subunit protein expression (**Fig. 5g, left panel**) (**lines 246-250**). This is very likely attributable to a long target protein half-life. In general, subunits of PP2A have been shown to be highly stable, e.g. C α subunits of PP2A are specifically protected from degradation by other cellular proteins (Yabe *et al.*, 2015, PMID: 26678046; Kong *et al.*, 2009, PMID: 19818709). Stable knockdown seemed not to be tolerated in those immune-competent cells.

To (ii): Immune-competent cells such as MDMs, U937 or THP-1s were proven to be unsuitable model to monitor infection upon inhibition of PP2A. We believe that the dephosphorylation of SAMHD1 by PP2A is opposed by other cellular functions of PP2A, the main contributing cause as why the detection window might only allow a minor effect upon infection in myeloid cells. Upon knockdown of single subunits (B55 α or C α) or triple knockdown of A α +B α +C α , as expected, phosphorylation at T592 increased in MDMs (see figure supporting rebuttal 2a and b, left panel). We observed, though, upregulation of the interferon pathway upon silencing of PP2A (as indicated by upregulation of STAT1). Recently, PP2A has been demonstrated to negatively regulate the IRF3 pathway upon stimulation or Sendai virus infection. IRF3

is then dephosphorylated with the consequence of suppressing VSV replication, as shown in Long *et al.* (Long *et al.*, 2014; PMID: 24726876). In fact, flaviviruses are known to utilize the negative regulatory role of PP2A for their own advantage and induce PP2A activity in order to block IFN-mediated restrictions (Duong *et al.*, 2005, PMID: 16306605; Duong *et al.*, 2004, PMID: 14699505). Additionally, PP2A depletion using siRNAs was shown to increase levels of pro-inflammatory cytokines *in vitro* (Cornell *et al.*, 2009, PMID: 19286927; Sun *et al.*, 2007, PMID: 17170118). We observed upregulation of STAT1 upon silencing of PP2A indicative of a host antiviral response. Consequently, subsequent infection experiments mirrored the antiviral state of the host cell, counteracting the effect of upregulated SAMHD1 phosphorylation in MDMs. Moreover, an antibody cocktail to block IFN (α -IFN α , α -IFN β and α -IFNAR) (see figure supporting rebuttal 2b, right panel) blocked STAT1 activation, but unfortunately affected the phosphorylation levels by an unknown mechanism, therefore prevented us from using this cocktail. Therefore, we determined sensing-competent cells as an unsuitable model to monitor infection.

To iii): Inhibition of PP2A (e.g. through chemical inhibition by OA) has been shown to affect many cellular processes; in particular, it inhibits CDK2 activity (Yan *et al.*, 1999; PMID: 10542219). As effective HIV-1 reverse transcription is known to be dependent on phosphorylation of the reverse transcriptase (RT) protein by CDK2 (Leng *et al.*, 2014; PMID: 24922574), inhibition of PP2A might inhibit effective RT and affect the stability of RT. Therefore, altered HIV-1 infectivity in MDMs mirroring the increase in SAMHD1 phosphorylation at T592 would be opposed by less efficient RT due to dephosphorylation of the RT protein. We believe that this point together with the antiviral state induced by IFN might be the major contributors explaining the minor effects we detect upon infection.

Additionally, we are not aware of any published synchronization protocol on primary CD4⁺ T cells, probably due to variability of individual donors. The most difficult phase in which to investigate events related to both entry into and exit from seems to be G₁, mainly because of the high degree of variability in G₁-phase progression (Jackman and O'Connor, 2001; PMID: 18228388), in particular in primary cells. Moreover, investigating G₁-sorted CD4⁺ T cell population upon infection is challenging as (i) cells will continue in the cell cycle and (ii) the sorted G₁ population will consist of a mixture of cells at the G₁/ S border and those at the M/ G₁ border reflecting differences in infection.

Nonetheless, we believe, PP2A is playing a role in relevant cells, as we do see binding of endogenous PP2A to endogenous SAMHD1 (**new Fig. 4d**). Additionally, in THP-1 KO SAMHD1 cells overexpressing FLAG-tagged SAMHD1, SAMHD1 does bind specifically to endogenous PP2A (figure supporting rebuttal 6a below). We also believe, that PP2A-B55 α holoenzymes play an important role at mitotic exit during the cell cycle in relevant cells, as in all mammalian cells (such as primary CD4⁺ T, THP-1 or HEK 293T cells) the PP2A-B55 α holoenzyme was screened as the only mammalian mitotic exit phosphatase responsible to remove phosphorylations on CDK substrates (Schmitz *et al.*, 2010; PMID: 20711181), such as shown here, for

SAMHD1. Therefore, we believe that the results on synchronization on HeLa 'Kyoto' cells, universally accepted in cell biology for cell cycle-experiments, together with the results on the chemically induced mitotic exit experiment prove that SAMHD1 is dephosphorylated during mitotic exit by PP2A-B55 α holoenzymes (**Fig. 1 and Fig. 6**).

Therefore, we focused our attention to prove that in cycling cells in the G₁ stage, SAMHD1 is an active enzyme (i) depleting dNTP pools, and (ii) can inhibit HIV-1. **New Fig. 7** demonstrates that SAMHD1 dephosphorylation correlates with a decrease in HIV-1 RT products upon G₁ entry.

- 9) *Figure 7C, D: The magnitude of the influence of OA treatment on viral replication step(s) is very modest. Authors may measure the luciferase activity in the parallel experiment. Previous reports demonstrated that human primary MDMs have only non-phosphorylated SAMHD1. However, authors show that MDMs (especially Donor 2) exhibit very high level of pT592 probably due to improper purification/differentiation of MDMs. Cell surface markers and cell-cycle profile could be analyzed by FACS analysis.*

As suggested by the reviewer, we now include cell surface marker and cell cycle-related marker of MDMs in **Supplementary Fig. 3**. Representative for six different MDM donors is depicted in **Supplementary Fig. 3a**: On average, we get purity of more than 95 % (CD14) and the cells express various macrophage markers that are not present in monocytes. In fact, as seen in **Supplementary Fig. 3b**: the levels of phosphorylated SAMHD1 in our MDMs are very low when e.g. compared to THP-1 cells and are also highly donor-dependent. Levels of T592 phosphorylation correlate with infectivity, further reinforcing the influence of SAMHD1 on HIV-1 infection efficiency (**Supplementary Fig. 3c**). In previous Fig. 7C, we presented a very long exposure time of the western blot in order to be able to compare to the low level of phosphorylation of donor 1 at that time. The long exposure time may have been misleading; therefore, we present now **Supplementary Fig. 3** to substantiate the general low level of phosphorylation in MDMs. As a note, other publications present phosphorylation of MDMs to be dependent on the use of cytokines, but also on the culture conditions and "pre"-stimulation by FBS (Mlcochova *et al.*, 2017, PMID: 28122869; Badia *et al.*, 2016, PMID: 27541004). Cribier *et al.*, for instance, present phosphorylation of MDMs upon GM-CSF stimulation as well (Cribier *et al.*, 2013; PMID: 23602554).

As pointed out by the reviewer, we intended to measure luciferase activity in parallel using a luciferase reporter virus: In contrast to RT products upon OA inhibition (as in previous Fig. 7C, D), luciferase expression is massively decreased upon OA treatment (see figure supporting rebuttal 3 below). Again, we believe, most likely an IFN-mediated block which acts past RT might affect luciferase. Additionally, indirect effects (OA is known to inhibit protein synthesis; e.g. Matias *et al.*, 1996, PMID:

8656452) or direct effects (CDK9/ CDK2 inhibition due to PP2A inhibition; Breuer *et al.*, 2012, PMID: 23140174) might be responsible for the massive drop in expression levels. Therefore, these additional effects prevented us to analyze luciferase gene expression.

The reviewer pointed out the modest magnitude of the influence of OA treatment on viral replication (measured by RT products). We currently believe, however, that the effects on viral replication through dephosphorylation of SAMHD1 by PP2A are opposed by several other cellular functions of PP2A. Besides the antiviral state induced by IFN (see answer to point 8 above), inhibition of PP2A (e.g. through chemical inhibition by OA) has been shown to affect many cellular processes, in particular, it suppresses CDK2 activity (Yan *et al.*, 1999; PMID: 10542219). As effective HIV-1 reverse transcription is known to be dependent on phosphorylation of the reverse transcriptase (RT) protein by CDK2 (Leng *et al.*, 2014; PMID: 24922574), inhibition of PP2A might inhibit effective RT and affect the stability of RT. Therefore, altered HIV-1 infectivity in MDMs mirroring the increase in SAMHD1 phosphorylation at T592 would be opposed by less efficient RT due to dephosphorylation of the RT protein. We believe, this point, together with the antiviral state induced by IFN, might be the major contributors of the minor effect we detect upon infection in MDMs. We determined sensing-competent cells, such as MDMs, as an unsuitable model to monitor infection upon inhibition of PP2A. Therefore, we excluded the results (previous Fig. 7C, D) from the revised version. Instead, we decided to set the focus of the manuscript on the effect of PP2A in proliferating cells and present now an alternative infection model on proliferating cells in **new Fig. 7**.

10) Figure 7E: The increased expression of PP2A-B55a in response to IFN treatments has not been mechanistically explored and could occur for a number of reasons, either direct or indirect. Does HIV infection induce PP2A expression? Moreover, it is not clear whether the increased expression of PP2A would affect the viral restriction in MDMs from the presented data. PP2A (wild-type, T592A, T592E)-transduced U937 cells could be useful for analyzing the function of IFN-induced PP2A in terms of HIV-1 infection. A more productive approach could be to more fully explore the exact role of the IFN-PP2A-SAMHD1 axis in HIV-1 replication.

We agree with the reviewer that we have not explored the mechanism behind the interferon inducibility of B55 α in MDMs. Nevertheless, we wanted to present this interesting result as this is the first time in PP2A biology reporting IFN-inducibility of one of the PP2A subunits. So far, exogenous stimuli have been reported to affect PP2A by directly regulating its activity, mostly through post-translational modifications of the holoenzyme. At this point, we cannot exclude that, additionally, post-translational modifications of PP2A might occur upon IFN treatment that alter its phosphatase activity to different extents, depending on the type of IFN applied. Therefore, future studies would be needed to characterize PP2A activity after IFN stimulation in more detail. Specific phosphatase activation through IFNs would allow

for an additional mechanism to control SAMHD1 function in different immunological and cellular contexts.

As pointed out in the response to specific point 8, we think that experiments on TPA-differentiated cell models (such as THP-1 or U937 cells overexpressing SAMHD1) will not reflect the situation of primary differentiated G₀/ G₁-cells and will not present a suitable cell model to answer the IFN-PP2A axis: (i) In our experience, type I and II IFNs do not induce any appreciable dephosphorylation of SAMHD1 at T592 in THP-1 cells in contrast to MDMs (see figure supporting rebuttal 6c). This is in agreement with published results as seen in Cribier *et al.* (Cribier *et al.*, 2013; PMID: 23602554): the authors demonstrate only minor dephosphorylation of SAMHD1 upon IFN treatment in THP-1 cells. (ii) Exogenous SAMHD1 is highly expressed upon TPA-induction in U937 (see figure supporting rebuttal 6d). In agreement with others (e.g. Lahouassa *et al.*, 2012; PMID: 22327569), TPA massively induces the expression of CMV-driven exogenous SAMHD1 constructs. We are not aware of any publication demonstrating dephosphorylation of SAMHD1 in an overexpression setting in U937 cells upon IFN addition. As there is massive overexpressed SAMHD1 present, we believe this system is unsuitable to observe dephosphorylation upon IFN.

We believe that HIV might have means to counteract PP2A action as additional mechanism (besides G₂/ M block) to avoid entering the restrictive G₁ phase. As discussed in **lines 476-484** of the revised manuscript, Vif has been shown to induce the proteasomal degradation of the B56 family of regulatory subunits of PP2A (Greenwood *et al.*, 2016; PMID: 27690223). We can verify their result and see reduction in B56 δ upon Vif expression. To further address the reviewer's concern, we analyzed whether the accessory proteins of HIV may have an influence on the expression level of B55 α (**see Supplementary Fig. 6**), however we could not observe any differences in expression compared to control (**see Supplementary Fig. 6**). Many viral proteins encoded by different virus families have been shown to influence PP2A by various means, for instance to change the activity of PP2A by either forming stable complexes with PP2A holoenzymes or with cellular proteins interacting with PP2A (Guergnon *et al.*, 2011; PMID: 21856415). We agree this is an interesting subject that needs further thorough and detailed investigation, but which would be out of the scope of this manuscript. We added to the discussion our hypothesis that proper control of mitotic exit events may be dependent on a phosphatase relay. This would suggest that antagonism of PP2A-B56 by HIV-1 Vif could possibly influence PP2A-B55 α action during mitotic exit (**lines 485-492**).

11) *Figure 7F: PP2A has multiple substrates so that endogenous one may not efficiently dephosphorylate SAMHD1 in cells. This can be analyzed by prior introduction of siRNA against B55a followed by IFN treatment. Also, it is plausible that an adaptor protein (ISG?) might modulate the interaction between PP2A and SAMHD1 as is often the case (e.g. Okamoto et al., Mol Cell. 2002 9:761-71).*

We silenced B55 α or performed triple knock-down of A α +B α +C α and analyzed how this may affect the IFN-induced dephosphorylation of SAMHD1 (figure supporting rebuttal 7). To be noted, as pointed out in response to point 8: highly efficient silencing of B55 α mRNA by RNAi (**Fig. 5g, right panel**) translates in apparent, but modest downregulation of B55 α subunit protein expression (**Fig. 5g, left panel**), probably due to the long target protein half-life. As expected, silencing of B55 α or PP2A-B55 α holoenzyme induced upregulation of phosphorylation (see figure supporting rebuttal 2a and b, left panel and figure supporting rebuttal 7). Secondly, as expected, upon IFN treatment, protein level of B55 α increased. We observed that protein level increased even in samples where RNAi had lowered the steady-state level of B55 α . However, this increase is mirrored by dephosphorylation at T592. We believe that it is very likely that the remaining B55 α protein level is responsible for the dephosphorylation at T592 upon IFN treatment. It is also possible that IFN treatment might regulate the activity of PP2A through e.g. post-translational modifications or upregulation of an interferon-regulated protein. We have added this point to the discussion (**lines 507-518**). Specific phosphatase activation through IFNs would allow for an additional mechanism to control SAMHD1 function. Therefore, future studies would be needed to characterize PP2A activity after IFN stimulation in more detail.

12) T592 is located adjacent to Vpx binding region. Does the interaction of PP2A with SAMHD1 affect the accessibility of Vpx to SAMHD1? Or, conversely, Vpx binding to SAMHD1 affects the PP2A-mediated dephosphorylation of SAMHD1? This could be tested in the presence of proteasomal inhibitors.

As presented in our answer to specific point 3, we now determined the PP2A-binding region within SAMHD1. We added this point to the discussion (**lines 426-435**). Binding of Vpx might not affect binding to PP2A as (i) the only two identified shared residues (609 and 622) for binding to both, Vpx or PP2A, were not defined as “key residues” for Vpx binding by Ahn *et al.* (Ahn *et al.*, 2012; PMID: 22362772) and (ii) more importantly, subcellular localization of both proteins may prevent interference by binding to SAMHD1. As B55 α is primarily localized to the cytoplasm (Lambrecht *et al.*, 2013, PMID: 23860660; Mo *et al.*, 2014, PMID: 25536081) and as the B regulatory subunits control the subcellular localization of the holoenzyme (Lambrecht *et al.*, 2013; PMID: 23860660), we speculate that during mitotic exit after nuclear envelope breakdown, PP2A-B55 α holoenzymes and SAMHD1 might get in contact and allow for rapid SAMHD1 dephosphorylation at residue T592 as observed in our study (**Supplementary Fig. 5b**). In contrast, SAMHD1 is first targeted by Vpx in the nucleus (Hofmann *et al.*, 2012, PMID: 22973040; Brandariz-Nuñez *et al.*, 2012, PMID: 22691373), whereas subsequent relocalization of the SAMHD1-Vpx complex to the cytoplasm may occur for proteasomal degradation of SAMHD1 (Brandariz-Nuñez *et al.*, 2012, PMID: 22691373, Laguette *et al.*, 2012, PMID: 22305291). The holoenzyme complexes are thought to interact with their substrates depending on the

spatial and temporal context and are regulated dynamically in response to environmental stimuli (Lambrecht *et al.*, 2013; PMID: 23860660).

13) Supplementary Figure 1C, right panel: Statistical analysis should be needed

New Supplementary Fig. 5c (former Supplementary Fig. 1C) got amended accordingly.

14) Supplementary Figure 2C: SAMHD1 blot looks very messy and should be amended.

New supplementary Fig. 4e (former Supplementary Fig. 2C) got amended accordingly.

15) This reviewer suggests that authors generate summarized cartoon of suggested mechanism.

As suggested, we added two cartoons to summarize the suggested mechanism in **new Fig. 8**.

Addition: Figures supporting rebuttal

Figure 1: SAMHD1 is dephosphorylated at T592 during mitotic exit - related to Fig. 6/ Supplementary Fig. 5.

HeLa 'Kyoto' cells were arrested in G_2/M phase using nocodazole and mitotic exit was induced chemically by adding flavopiridol (FP) in the presence of MG-132. Cells were harvested at the indicated time points over a period of 18 min. Whole-cell lysates were analyzed by immunoblotting using antibodies specific to the indicated proteins. Phosphorylated SAMHD1 (= SAMHD1 pT592; signal marked with asterisks) was identified by comparing mitotic (= 0/ 18 min) to asynchronously (= A) growing cells and correlating the SAMHD1 pT592-band with total SAMHD1. Data shown are representative of three independent experiments.

Figure 2: PP2A silencing influences SAMHD1 phosphorylation at T592 and increases IFN signaling in monocyte-derived macrophages (MDMs) – related to Fig. 5.

(a) siRNA-mediated silencing of PP2A increases SAMHD1 T592 phosphorylation and STAT1 levels in MDMs. MDMs were transfected twice with control siRNA or a siRNA specifically targeting PP2A B55α/ Cα/ Aα+Bα+Cα (= Triple) subunits. 48 h post-transfection, cells were harvested and whole-cell lysates were analyzed by using antibodies specific to the indicated proteins.

(b) Blocking of IFN signaling influences SAMHD1 T592 phosphorylation status. 1 h prior to siRNA transfection, MDMs were either incubated with different α -IFN antibodies (α -IFN α , α -IFN β and α -IFNAR, 1 μ g/ mL each) or left untreated (= w/o). Subsequently, MDMs were transfected twice with control siRNA or a siRNA specifically targeting PP2A B55α/ Cα/ Aα+Bα+Cα (= Triple) subunits. With each siRNA transfection and medium change, α -IFN antibodies were freshly added. 5 d post-transfection, cells were harvested and whole-cell lysates were analyzed by using antibodies specific to the indicated proteins. Data shown are representative of two experiments analyzed (0.5 and 1 μ g/ mL α -IFN antibodies).

Figure 3: PP2A inhibition by OA reduces luciferase expression after HIV-1 infection of MDMs-related to former Fig. 7C/ D.

(a) Before infection, MDMs were pre-incubated for 2 h with DMSO (as control) or 10 nM OA. After OA treatment, MDMs were infected with VSV-G-pseudotyped HIV-1-luciferase reporter virus (MOI 10). For control purposes, the used virus was either heat-inactivated or the RT inhibitor azidothymidine (AZT) added at the time of infection. After 24 h, infection efficiency was determined by measuring luciferase activity. Graphs represent the mean \pm SD of technical triplicates measured for each sample. Data shown are representative of four donors analyzed.

(b) Relative cell viability of each donor infected in (a). Graphs represent the mean \pm SD of technical triplicates measured for each sample. Data shown are representative of four donors analyzed.

Figure 4: Influence of C-terminal basic amino acids on interaction of SAMHD1 with PP2A-B55 α holoenzymes - related to Fig. 4.

(a) Schematic of basic amino acids in the C-terminal region of human SAMHD1. Basic residues are highlighted in black, while introduction of non-basic alanines at the corresponding positions are highlighted in gray.

(b) Expression level of N-terminally FLAG-tagged human SAMHD1 or mutants with substituted basic residues (= R/ K to A) in the C-terminal region. HEK 293T cells were transfected with constructs expressing N-terminally FLAG-tagged human SAMHD1 or mutants. Empty vector was transfected as a negative control. 48 h post-transfection, cells were harvested and lysed. Proteins were analyzed by immunoblotting using antibodies specific to the indicated proteins.

(c) Influence of C-terminal basic amino acids on interaction of SAMHD1 with PP2A-B55 α holoenzymes. HEK 293T cells were transfected with constructs expressing N-terminally FLAG-tagged human SAMHD1 or mutants with substituted basic residues (= R/ K to A). Empty vector was transfected as a negative control. 48 h post-transfection, cells were harvested, lysed and CoIP performed using anti-FLAG-coated agarose beads. Proteins were analyzed by immunoblotting using

antibodies specific to the indicated proteins. Data shown are representative of two independent experiments.

a

b

Figure 5: Inhibition of PP2A by okadaic acid (OA) increases SAMHD1 phosphorylation at residue T592 in cycling HeLa ‘Kyoto’ cells - related to Fig. 5d.

HeLa ‘Kyoto’ cells were treated with the phosphatase inhibitor OA (2 nM) and harvested at different time points. Whole-cell lysates were analyzed by immunoblotting using antibodies specific to the indicated proteins, for instance, **(a)** α-SAMHD1 (10 s exposure) and **(b)** α-Cyclin B1 (3 min exposure). Data shown are uncropped scans of autoradiography films shown in Fig. 5d.

Figure 6: Phosphorylation level at T592 in SAMHD1 in cycling monocytic cells upon PP2A inhibition or IFN stimulation.

(a) SAMHD1 interacts with endogenous PP2A A/ B55 α / C subunit in cycling THP-1 cells. THP-1 SAMHD1 KO cells were stably transduced with N-terminally FLAG-tagged SAMHD1 or empty vector as a negative control. Cells were harvested, lysed and CoIP performed using anti-FLAG-coated agarose beads. Proteins were analyzed by immunoblotting using antibodies specific to the indicated proteins.

(b) Inhibition of PP2A by okadaic acid (OA) in THP-1 cells does not increase SAMHD1 phosphorylation at residue T592. Cycling THP-1 cells (-TPA) were treated with the phosphatase inhibitor OA (20 nM) and harvested at different time points. Whole-cell lysates were analyzed by immunoblotting using antibodies specific to the indicated proteins. Data shown are representative of three independent experiments with variant OA concentration.

(c) Type I and II IFNs do not induce dephosphorylation of SAMHD1 at T592 in THP-1 cells. Whole-cell lysates of THP-1 cells after treatment with different IFNs (IFN α , IFN β , IFN γ ; 1000 U/ mL each) for 24 h and TPA treatment (-/ +TPA) were analyzed by immunoblotting using antibodies specific to the indicated proteins.

(d) TPA-induced differentiation induces expression of exogenous SAMHD1 in monocytic U937 cells. U937 cells were stably transduced N-terminally FLAG-tagged SAMHD1 or empty vector (Ctr). After puromycin selection, cells were differentiated using different TPA concentrations (25, 50, 100 ng/ mL) overnight and whole-cell lysates were analyzed by immunoblotting using antibodies specific to the indicated proteins.

Figure 7: siRNA-mediated silencing of PP2A does not restore SAMHD1 T592 phosphorylation after IFN stimulation in MDMs.

MDMs were transfected with control siRNA or a siRNA specifically targeting PP2A B55α subunit/PP2A-B55α holoenzyme (Aα+B55α+Cα). 48 h post-transfection, cells were either left untreated (mock) or stimulated with IFNα/ β (1000 U/ mL). After 24 h, MDMs were harvested and whole-cell lysates were analyzed by using antibodies specific to the indicated proteins. Data shown represent one of two donors analyzed.

Addition: References supporting rebuttal

- Ahn, J. et al. HIV/simian immunodeficiency virus (SIV) accessory virulence factor Vpx loads the host cell restriction factor SAMHD1 onto the E3 ubiquitin ligase complex CRL4DCAF1. *J. Biol. Chem.* **287**, 12550–12558 (2012).
- Badia, R. et al. The G1/S Specific Cyclin D2 Is a Regulator of HIV-1 Restriction in Non-proliferating Cells. *PLoS Pathog.* **12**, e1005829 (2016).
- Bhattacharya, S. et al. SKP2 associates with p130 and accelerates p130 ubiquitylation and degradation in human cells. *Oncogene* **22**, 2443–2451 (2003).
- Brandariz-Nuñez, A. et al. Role of SAMHD1 nuclear localization in restriction of HIV-1 and SIVmac. *Retrovirology* **9**, 49 (2012).
- Breuer, D. et al. CDK2 regulates HIV-1 transcription by phosphorylation of CDK9 on serine 90. *Retrovirology* **9**, 94 (2012).
- Cornell, T. T. et al. Ceramide-dependent PP2A regulation of TNF α -induced IL-8 production in respiratory epithelial cells. *Am. J. Physiol. Lung Cell. Mol. Physiol.* **296**, L849-56 (2009).
- Cribier, A., Descours, B., Valadao, A. L. C., Laguette, N. & Benkirane, M. Phosphorylation of SAMHD1 by cyclin A2/CDK1 regulates its restriction activity toward HIV-1. *Cell Rep.* **3**, 1036–1043 (2013).
- Cundell, M. J. et al. The BEG (PP2A-B55/ENSA/Greatwall) pathway ensures cytokinesis follows chromosome separation. *Mol. Cell* **52**, 393–405 (2013).
- Cundell, M. J. et al. A PP2A-B55 recognition signal controls substrate dephosphorylation kinetics during mitotic exit. *J. Cell Biol.* **214**, 539–554 (2016).
- Duong, F. H. T., Filipowicz, M., Tripodi, M., La Monica, N. & Heim, M. H. Hepatitis C virus inhibits interferon signaling through up-regulation of protein phosphatase 2A. *Gastroenterology* **126**, 263–277 (2004).
- Duong, F. H. T. et al. Upregulation of protein phosphatase 2Ac by hepatitis C virus modulates NS3 helicase activity through inhibition of protein arginine methyltransferase 1. *J. Virol.* **79**, 15342–15350 (2005).
- Greenwood, E. J. et al. Temporal proteomic analysis of HIV infection reveals remodelling of the host phosphoproteome by lentiviral Vif variants. *eLife* **5** (2016).
- Guergnon, J. et al. PP2A targeting by viral proteins. A widespread biological strategy from DNA/RNA tumor viruses to HIV-1. *Biochim. Biophys. Acta* **1812**, 1498–1507 (2011).
- Hofmann, H. et al. The Vpx lentiviral accessory protein targets SAMHD1 for degradation in the nucleus. *J. Virol.* **86**, 12552–12560 (2012).
- Lahouassa, H. et al. SAMHD1 restricts the replication of human immunodeficiency virus type 1 by depleting the intracellular pool of deoxynucleoside triphosphates. *Nat. Immunol.* **13**, 223–228 (2012).
- Jackman, J. & O'Connor, P. M. Methods for synchronizing cells at specific stages of the cell cycle. *Curr. Protoc. Cell Biol.* **Chapter 8**, Unit 8.3 (2001).
- Janssens, V. et al. Identification and functional analysis of two Ca²⁺-binding EF-hand motifs in the B⁰/PR72 subunit of protein phosphatase 2A. *J. Biol. Chem.* **278**, 10697–10706 (2003).
- Kong, M., Ditsworth, D., Lindsten, T. & Thompson, C. B. Alpha4 is an essential regulator of PP2A phosphatase activity. *Mol. Cell* **36**, 51–60 (2009).
- Laguette, N. et al. Evolutionary and functional analyses of the interaction between the myeloid restriction factor SAMHD1 and the lentiviral Vpx protein. *Cell Host Microbe* **11**, 205–217 (2012).

- Lambrecht, C., Haesen, D., Sents, W., Ivanova, E. & Janssens, V. Structure, regulation, and pharmacological modulation of PP2A phosphatases. *Methods Mol. Biol.* **1053**, 283–305 (2013).
- Leng, J. et al. A cell-intrinsic inhibitor of HIV-1 reverse transcription in CD4(+) T cells from elite controllers. *Cell Host Microbe* **15**, 717–728 (2014).
- Leulliot, N. et al. Structure of protein phosphatase methyltransferase 1 (PPM1), a leucine carboxyl methyltransferase involved in the regulation of protein phosphatase 2A activity. *J. Biol. Chem.* **279**, 8351–8358 (2004).
- Long, L. et al. Recruitment of phosphatase PP2A by RACK1 adaptor protein deactivates transcription factor IRF3 and limits type I interferon signaling. *Immunity* **40**, 515–529 (2014).
- Matias, W. G., Bonini, M. & Creppy, E. E. Inhibition of protein synthesis in a cell-free system and Vero cells by okadaic acid, a diarrhetic shellfish toxin. *J. Toxicol. Environ. Health* **48**, 309–317 (1996).
- McCourt, P., Gallo-Ebert, C., Gonghong, Y., Jiang, Y. & Nickels, J. T. PP2A(Cdc55) regulates G1 cyclin stability. *Cell Cycle* **12**, 1201–1210 (2013).
- Mlcochova, P. et al. A G1-like state allows HIV-1 to bypass SAMHD1 restriction in macrophages. *EMBO J.* **36**, 604–616 (2017).
- Mo, S.-T. et al. Visualization of subunit interactions and ternary complexes of protein phosphatase 2A in mammalian cells. *PLoS One* **9**, e116074 (2014).
- Schmitz, M. H. A. et al. Live-cell imaging RNAi screen identifies PP2A-B55alpha and importin-beta1 as key mitotic exit regulators in human cells. *Nat. Cell Biol.* **12**, 886–893 (2010).
- Schwefel, D. et al. Structural basis of lentiviral subversion of a cellular protein degradation pathway. *Nature* **505**, 234–238 (2014).
- Seshacharyulu, P., Pandey, P., Datta, K. & Batra, S. K. Phosphatase: PP2A structural importance, regulation and its aberrant expression in cancer. *Cancer Lett.* **335**, 9–18 (2013).
- Sun, L. et al. Tristetraprolin (TTP)-14-3-3 complex formation protects TTP from dephosphorylation by protein phosphatase 2a and stabilizes tumor necrosis factor-alpha mRNA. *J. Biol. Chem.* **282**, 3766–3777 (2007).
- Traore, K. et al. Signal transduction of phorbol 12-myristate 13-acetate (PMA)-induced growth inhibition of human monocytic leukemia THP-1 cells is reactive oxygen dependent. *Leuk. Res.* **29**, 863–879 (2005).
- Yabe, R. et al. Protein Phosphatase Methyl-Esterase PME-1 Protects Protein Phosphatase 2A from Ubiquitin/Proteasome Degradation. *PLoS One* **10**, e0145226 (2015).
- Yan, Y. & Mumby, M. C. Distinct roles for PP1 and PP2A in phosphorylation of the retinoblastoma protein. PP2a regulates the activities of G(1) cyclin-dependent kinases. *J. Biol. Chem.* **274**, 31917–31924 (1999).

Reviewers' comments:

Reviewer expertise:

Reviewer #2: PP2A, cell cycle

Reviewer #3: HIV, SAMHD1

Reviewer #4: HIV, SAMHD1

Reviewer #2 (Remarks to the Author):

I am very impressed with the care with which the authors have addressed the concerns of the reviewers. These efforts included a number of new confirmatory experiments and discussion of various aspects of PP2A-B55 regulation that are central to an appreciation of how the events studied by the authors relate to control of the cell cycle. As a result, the results are more persuasive and the story line is much more coherent. I thus urge publication in Nature Communications, and I do not recommend requirements for further experimentation.

One disappointing aspect of the revised manuscript is the Discussion, which goes on seemingly forever. At a minimum, the Discussion would be improved by adding subheadings that help guide the reader through this morass. Likely the Discussion could benefit as well from a compression that removes 10-20% of the verbiage, much of which presently is repetitious. However, one slight addition to the Discussion would also be useful: the mechanism that explains how phosphorylated Endosulfine inhibits PP2A-B55 and how PP2A-B55 inactivates phosphorylated Endosulfine has recently been described (eLife 2017;6:e24665). This information would help clarify why PP2A-B55's action on SAMHD1 is inhibited through M phase but then gets turned on during M phase exit.

Reviewer #3 (Remarks to the Author):

Although I still have a concern about the pathophysiological significance of SAMHD-1 dephosphorylation by PP2A in HIV infection, overall the manuscript is improved, appears to be technically sound and with new data adequately addressed the concerns of this reviewer. All other minor issues were also addressed appropriately so that there are no further remaining criticisms in this stage.

Reviewer #4 (Remarks to the Author):

The authors made a significant effort to address the issues raised by reviewer #1, for the most part successfully so. The data suggesting that PP2A is the cellular phosphatase responsible for the reversible dephosphorylation of SAMHD1 are indeed convincing. Less convincing are the issues raised in points 7 & 8 of the rebuttal. Concerning point 7: The authors indicate that the results shown in Fig. 5g (which is one of the few experiments done in actual HIV target cells) represents data from one of three donors. Assuming they show the best data set and the effect obtained with the other two donors is even less pronounced,

this experiment does not make a compelling case for a significant role of PP2A in regulating SAMHD1 phosphorylation in human macrophages. Concerning point 8: The authors decided to replace results obtained with human macrophages by results obtained in synchronized HeLa 'Kyoto' cells. Even so, the results shown in the new Fig. 7b are not very strong; they may be statistically significant, but I question their biological relevance since the effect is less than 2-fold and the infection was done with a whopping MOI of 3. With regard to the new figures 1c and 6b in which the authors attempt to correlate SAMHD1 phosphorylation status with the ability to lower dNTP levels, I would like to point out that two previous studies failed to see such a correlation (Welbourn et al., *Virology* 488:271-277 [2016]; Wang et al., *Virology* 487:273-284 [2016]). At a minimum, the authors should discuss this apparent discrepancy and put their results into context with published literature. Regarding Fig. 1c, Welbourn et al reported that HeLa cells contain high levels of dNTPs and effects of SAMHD1 on dNTP levels in these cells were only observed after inhibition of ribonucleotide reductase. It is possible that HeLa 'Kyoto' cells are special and differ from 'normal' HeLa cells. However, I could not find a description of the special features of these cells anywhere in the supplied information. If there is something special about HeLa 'Kyoto' cells, this should be clearly stated and referenced. Also, the authors should comment on how dNTP levels observed in Fig. 1c at the 11.5-hour time point compare to dNTP levels in differentiated MDM. Unless HeLa 'Kyoto' cells are very different from 'normal' HeLa cells, I would predict that dNTP levels in MDM are lower (yet permit HIV replication).

Minor comments:

Lines 169-171: The conclusions from Fig. 4a that SAMHD1 was only enriched after pull-down of the PP2A-B55alpha holoenzyme is not valid since GST-IP of B56gamma and B56delta did not work and, in addition, SAMHD1 levels are lower in some of these samples. This statement should be softened.

Fig. 4d: In figs 4b and 4c, the +/- indicates the presence or absence of nFlag-SAMHD1 (as indicated in the figure label). However, the +/- label in Fig. 4c is undefined. It can't be +/- SAMHD1 since the experiment analyzes endogenous SAMHD1 in THP-1 cells.

We would like to thank the reviewers for their valuable feedback. We have made revisions as pointed out below and included additional experimental studies to address the concerns raised by reviewer #4 (**Fig. 7; Supplementary Fig. 3e, f; Supplementary Fig. 6a, b and c; Supplementary Fig. 7**). A point-by-point response is provided below.

Reviewer #2 (Remarks to the Author):

I am very impressed with the care with which the authors have addressed the concerns of the reviewers. These efforts included a number of new confirmatory experiments and discussion of various aspects of PP2A-B55 regulation that are central to an appreciation of how the events studied by the authors relate to control of the cell cycle. As a result, the results are more persuasive and the story line is much more coherent. I thus urge publication in Nature Communications, and I do not recommend requirements for further experimentation.

One disappointing aspect of the revised manuscript is the Discussion, which goes on seemingly forever. At a minimum, the Discussion would be improved by adding subheadings that help guide the reader through this morass. Likely the Discussion could benefit as well from a compression that removes 10-20% of the verbiage, much of which presently is repetitious. However, one slight addition to the Discussion would also be useful: the mechanism that explains how phosphorylated Endosulfine inhibits PP2A-B55 and how PP2A-B55 inactivates phosphorylated Endosulfine has recently been described (eLife 2017;6:e24665). This information would help clarify why PP2A-B55's action on SAMHD1 is inhibited through M phase but then gets turned on during M phase exit.

We thank the reviewer for highlighting our results as persuasive and the story line coherent. We revised the discussion as suggested, added subheadings, reduced repetitious text passages and included the valuable addition as pointed out (**discussion page 16-17**).

Reviewer #3 (Remarks to the Author):

Although I still have a concern about the pathophysiological significance of SAMHD-1 dephosphorylation by PP2A in HIV infection, overall the manuscript is improved, appears to be technically sound and with new data adequately addressed the concerns of this reviewer. All other minor issues were also addressed appropriately so that there are no further remaining criticisms in this stage.

We are glad to hear that the reviewer feels that the inclusion of new data were adequate and technically sound. Even so the reviewer had no remaining criticism, we wanted to point out that we now provide relevance for SAMHD1 dephosphorylation in primary CD4⁺ T cells, primary targets for HIV infection (**see new Fig. 7**).

Reviewer #4 (Remarks to the Author):

The authors made a significant effort to address the issues raised by reviewer #1, for the most part successfully so. The data suggesting that PP2A is the cellular phosphatase responsible for the reversible dephosphorylation of SAMHD1 are indeed convincing.

Less convincing are the issues raised in points 7 & 8 of the rebuttal. Concerning point 7: The authors indicate that the results shown in Fig. 5g (which is one of the few experiments done in actual HIV target cells) represents data from one of three donors. Assuming they show the best data set and the effect obtained with the other two donors is even less pronounced, this experiment does not make a compelling case for a significant role of PP2A in regulating SAMHD1 phosphorylation in human macrophages.

In order to reinforce the role of PP2A in regulating SAMHD1 phosphorylation in macrophages, we added two new figures (**new Supplementary Fig. 3e, f**). Okadaic acid is considered a specific inhibitor for PP2A in low nM concentration (2-3 log difference to PP1; Lambrecht *et al.*, 2013, PMID: 23860660; Cohen *et al.*, 1989, PMID: 2546812). As 2 nM okadaic acid treatment is considered to inhibit completely and specifically PP2A, we used this concentration in the time-course experiment in Fig. 5f. **New Supplementary Fig. 3e** demonstrates a rapid increase in phosphorylation at T592 only after 3 hours of 10 nM okadaic acid treatment in four more macrophage donors. This figure also demonstrates that despite high donor variability (variant MCM2 expression and phosphorylation status at time point 0) SAMHD1 phosphorylation at T592 increased upon inhibition of PP2A.

Since we observed that efficient silencing of B55 α mRNA (Fig. 5g, right panel) translates only in a modest downregulation of B55 α subunit protein expression at 52 hours (Fig. 5g, left panel), presumably attributable to a long target protein half-life (Yabe *et al.*, 2015, PMID: 26678046; Kong *et al.*, 2009, PMID: 19818709), we transfected siRNAs targeting B55 α or C α twice and monitored protein levels over a time course of 5 days (**new Supplementary Fig. 3f**). Efficient silencing of PP2A at day 5 results in a remarkable increase in phosphorylation at T592 (**new Supplementary Fig. 3f**) and demonstrates the direct involvement of PP2A in macrophages.

In this rebuttal, we include two more experiments (two more different macrophage donors) representative for Fig. 5g (see **Figures supporting rebuttal 1**, below) demonstrating that specific silencing of B55 α leads to increase of SAMHD1 T592 phosphorylation in macrophages, indicating that PP2A-B55 α holoenzymes are mediating SAMHD1 dephosphorylation at this specific residue in primary cells.

Concerning point 8: The authors decided to replace results obtained with human macrophages by results obtained in synchronized HeLa 'Kyoto' cells. Even so, the results shown in the new Fig. 7b are not very strong; they may be statistically significant, but I

question their biological relevance since the effect is less than 2-fold and the infection was done with a whopping MOI of 3.

We agree with the reviewer that primary target cells of HIV-1 will prove biological relevance. We now present results on infection of synchronized activated CD4⁺ T cells in **new Fig. 7** (four different donors) and moved the HeLa results to **new Supplementary Fig. 6**. We optimized a synchronization protocol for activated CD4⁺ T cells using nocodazole which arrests the cell cycle at the prometaphase of mitosis. We determined in time-course experiments (not shown) that - upon removal of nocodazole - the cells exit into G₁ within 6 hours as indicated by PI stain (not shown), degradation of cyclin B1 and reduced phosphorylation at T592 in SAMHD1 than compared to cells in mitosis (**new Fig. 7f**). Importantly, the time of removal or continuation of nocodazole was chosen in such a way that sufficient DNA copies could be detected and coincided with specific cell cycle-phases. Upon removal of nocodazole at time point 18 h, cells could progress to G₁ (**new Fig. 7**, condition “**inf**”), whereas upon continuation, cells remain in mitosis (**new Fig. 7**, condition “**inf +N**”).

To optimize the infection experiment, we determined that early and late RT copies could be sufficiently detected at 12-16 hours post infection (data not shown), which is in agreement with Mohammadi *et al.*, 2013 (Fig. 1a or Suppl. Fig. S2; PMID: 23382686), who detected the peak of early and late RT products around these time points in T cell lines. The bulk of activated CD4⁺ T cells consists of a high proportion of (mainly refractory) G₀/ G₁ cells (*in vivo* >95% Corneau *et al.*, 2017, PMID: 27997758; *in vitro* (dependent upon timing and stimulus) 48-91% Krummel *et al.*, 1996, PMID: 8676074; own observations >83% as measured by PI stain, not shown). Even after synchronization by nocodazole, a high proportion of CD4⁺ T cells remain in (potentially refractory) G₁ phase. Therefore, we titrated the amount of virus that gave us sufficient signal for early and late RT (data not shown) and led to degradation of SAMHD1 in the bulk of CD4⁺ T cells upon incorporation of Vpx (**new Fig. 7f**).

As a result (**new Fig. 7**), we detected higher early and late HIV-1 RT products in mitotic cells, which displayed high SAMHD1 pT592-levels (**inf +N**), than compared to activated CD4⁺ T cells in the G₁ phase (**inf**) (**Fig. 7b, d**). As seen before in HeLa cells (**new Supplementary Fig. 6**), infection with HIV-1 in the presence of Vpx led to SAMHD1 degradation (**Fig. 7f**) and a significantly less pronounced difference in HIV-1 products (**Fig. 7c, e**). Therefore, we conclude that the observed reduction of HIV-1 RT products upon G₁ entry in cycling CD4⁺ T cells is dependent on the presence of dephosphorylated SAMHD1 active against HIV-1.

The results in primary CD4⁺ T cells (**new Fig. 7**) parallels the HeLa results (**new Supplementary Fig. 6**). We would like to point out that in synchronized HeLa cells we measured RT products early, 3 hours upon infection, to prevent cells from

progressing further in the cell cycle coinciding with specific cell cycle-phases (to some extent refractory to infection). Luciferase gene-expression read-out at 24 hpi in unsynchronized HeLa cells was well detectable with MOI of 0.5 (data not shown). We chose the amount of virus (i) resulting in sufficient DNA copies detected in G₁ stage, and (ii) displaying comparable copy numbers of RT products between experiments (+/- Vpx). For the synchronized HeLa cells, we chose 1.5 MOI (Vpx-carrying virus) and 3 MOI (non-Vpx carrying virus) (see **new Supplementary Fig. 6d-i**).

With regard to the new figures 1c and 6b in which the authors attempt to correlate SAMHD1 phosphorylation status with the ability to lower dNTP levels, I would like to point out that two previous studies failed to see such a correlation (Welbourn et al., Virology 488:271-277 [2016]; Wang et al., Virology 487:273-284 [2016]). At a minimum, the authors should discuss this apparent discrepancy and put their results into context with published literature.

We agree with the reviewer that (i) phosphorylation status of SAMHD1 may not be causally correlated with dNTP levels and (ii) that mitotic exit may trigger additional mechanisms of restriction by SAMHD1. We attempted to highlight that the interpretation of our data does not allow concluding whether the relationship is coincidental or causative. Our wording in the previously submitted version may have led to this misunderstanding, therefore we have thoroughly revised the manuscript to avoid misinterpretation of the data and extended the discussion on this point including the suggested publications (lines 80-82; 383-386; 420-426; 511-527; moved previous Fig. 6b to Supplementary Fig. 5c).

Regarding Fig. 1c, Welbourn et al reported that HeLa cells contain high levels of dNTPs and effects of SAMHD1 on dNTP levels in these cells were only observed after inhibition of ribonucleotide reductase. It is possible that HeLa 'Kyoto' cells are special and differ from 'normal' HeLa cells. However, I could not find a description of the special features of these cells anywhere in the supplied information. If there is something special about HeLa 'Kyoto' cells, this should be clearly stated and referenced.

HeLa 'Kyoto' cells are an isolate from HeLa cells (Resource Identification: RRID:CVCL_1922), particularly suitable for imaging (Schmitz et al., 2010; PMID: 20711181). The cells do not differ from regular HeLa cells. We confirmed the authentication of the cells as HeLa cells by examining genetic characteristics through PCR-single-locus-technology. We extended the information on this cell clone in the Supplementary Experimental Procedures and therefore minimized the information in the main text to "HeLa". In **new Supplementary Fig. 6a-c**, we measured the absolute levels of dNTP/10⁶ HeLa "Kyoto" cells. In synchronized HeLa "Kyoto" cells the dATP levels in different cell cycle phases (on average 89.4, 138.02, 17.97 pmol dATP/10⁶ cells in S, G₂/M, G₁, respectively) are comparable to non-synchronized HeLa cells reported by Welbourn et al. (around 25 to 45 pmol dATP/10⁶ cells). We speculate about the possibility of additional mechanism of restriction that are triggered upon mitotic exit and correlate with dephosphorylated SAMHD1 at T592

(see discussion page 18-19). This would agree with Welbourn *et al.* providing evidence for a dNTPase-independent function. We speculate that unsynchronized HeLa cells where dNTP levels were artificially lowered by inhibiting the RNR would parallel the situation in our experiment of synchronized HeLa cells where the G₁ stage provides a state of low dNTP levels (page 18-19).

Also, the authors should comment on how dNTP levels observed in Fig. 1c at the 11.5-hour time point compare to dNTP levels in differentiated MDM. Unless HeLa 'Kyoto' cells are very different from 'normal' HeLa cells, I would predict that dNTP levels in MDM are lower (yet permit HIV replication).

As suggested by the reviewer, we analyzed the dNTP levels of primary MDMs and additionally in resting CD4⁺ T cells (**new Supplementary Fig. 7**) and compared the levels to the G₁ phase in synchronized HeLa cells (**new Supplementary Fig. 6a-c**). We suggest that the influence of SAMHD1 on restrictive activity (**new Fig. 7**) is correlated with dephosphorylation of SAMHD1 at the mitotic exit (**Fig. 6a**). However, the exact mechanism of restriction still needs to be determined. The mitotic exit leads to a drop in dNTP levels in G₁ and dCTP seems to be the rate-limiting deoxynucleotide in the G₁ phase (**new Supplementary Fig. 6a-c**). When taking into account the differences in cell volumes of the various cell types, HeLa cells in G₁ phase contain ~21–35-fold and ~7–11-fold more dCTP than macrophages and resting CD4⁺ T cells, respectively (**new Supplementary Fig. 7d**), questioning whether the observed dNTP levels in G₁ may influence the critical threshold required for reverse transcription. When taking into account the range of K_m and K_d values of HIV-1 RT (as discussed on page 18-19), this suggests that the lowest measured level in G₁ phase that would be rate-limiting (1.09 – 1.8 μM dCTP) lies still above the general K_m values of HIV-1 RT and might not significantly affect the replication kinetics. As this would need further investigation, dNTP-independent influence of SAMHD1 during G₁ phase cannot be excluded.

Minor comments:

Lines 169-171: The conclusions from Fig. 4a that SAMHD1 was only enriched after pull-down of the PP2A-B55alpha holoenzyme is not valid since GST-IP of B56gamma and B56delta did not work and, in addition, SAMHD1 levels are lower in some of these samples. This statement should be softened.

As suggested, the statement got amended (lines 177-183). Indeed, we cannot completely exclude binding of other subtypes (additionally those that were not tested in this assay, as there are 23 isoforms). It is likely that other subtypes may interact with SAMHD1 with lower affinity and target other phosphorylation sites. But we would like to emphasize that we validated B55α by *in vitro*-dephosphorylation experiments to prove that B55α is the responsible B-type subunit leading to dephosphorylation at

the specific site T592 (Fig. 5c and Supplementary Fig. 4e) and we determined the recognition site in SAMHD1 shared only by B55 substrates (Fig. 4e and f). Specifically B55 α holoenzymes play a key regulatory role during mitotic exit (Schmitz *et al.*, 2010; PMID: 20711181). We discussed this point in line 439.

Fig. 4d: In figs 4b and 4c, the +/- indicates the presence or absence of nFlag-SAMHD1 (as indicated in the figure label). However, the +/- label in Fig. 4c is undefined. It can't be +/- SAMHD1 since the experiment analyzes endogenous SAMHD1 in THP-1 cells.

The incorrect label got changed.

Figures supporting Rebuttal

Figure 1: Silencing of PP2A B55 α subunit increases SAMHD1 phosphorylation at T592 in monocyte-derived macrophages (MDMs) – related to Fig. 5.

(a)+(b) siRNA-mediated silencing of PP2A increases SAMHD1 T592 phosphorylation in MDMs. MDMs were transfected twice with control siRNA or a siRNA specifically targeting PP2A B55 α subunit. 48 h **(a)** or 5 d **(b)** post-transfection, cells were harvested and whole-cell lysates were analyzed by using antibodies specific to the indicated proteins. Results show two different donors analyzed.

References

Cohen, P., Klumpp, S. & Schelling, D. L. An improved procedure for identifying and quantitating protein phosphatases in mammalian tissues. *FEBS Lett.* 250, 596–600 (1989).

Corneau, A. et al. Comprehensive Mass Cytometry Analysis of Cell Cycle, Activation, and Coinhibitory Receptors Expression in CD4 T Cells from Healthy and HIV-Infected Individuals. *Cytometry B Clin. Cytom.* 92, 21–32 (2017).

Kong, M., Ditsworth, D., Lindsten, T. & Thompson, C. B. Alpha4 is an essential regulator of PP2A phosphatase activity. *Mol. Cell* 36, 51–60 (2009).

Krummel, M. F. & Allison, J. P. CTLA-4 engagement inhibits IL-2 accumulation and cell cycle progression upon activation of resting T cells. *J. Exp. Med.* 183, 2533–2540 (1996).

Lambrecht, C., Haesen, D., Sents, W., Ivanova, E. & Janssens, V. Structure, regulation, and pharmacological modulation of PP2A phosphatases. *Methods Mol. Biol.* 1053, 283–305 (2013).

Mohammadi, P. et al. 24 hours in the life of HIV-1 in a T cell line. *PLoS Pathog.* 9, e1003161 (2013).

Schmitz, M. H. A. et al. Live-cell imaging RNAi screen identifies PP2A-B55alpha and importin-beta1 as key mitotic exit regulators in human cells. *Nat. Cell Biol.* 12, 886–893 (2010).

Welbourn, S. & Strebel, K. Low dNTP levels are necessary but may not be sufficient for lentiviral restriction by SAMHD1. *Virology* 488, 271–277 (2016).

Yabe, R. et al. Protein Phosphatase Methyl-Esterase PME-1 Protects Protein Phosphatase 2A from Ubiquitin/Proteasome Degradation. *PloS one* 10, e0145226 (2015).

REVIEWERS' COMMENTS:

Reviewer #4 (Remarks to the Author):

I think I have tortured the authors enough. They have done a remarkable job to address my criticisms and I have no further suggestions for improvement.